# Ensa controls S-phase length by modulating Treslin levels

Sophie Charrasse[1], Aicha Gharbi-Ayachi[1], Andrew Burgess [2,3], Jorge Vera[1], Khaled Hached[1], Peggy Raynaud[1], Etienne Schwob[4], Thierry Lorca[1] & Anna Castro[1]

The Greatwall/Ensa/PP2A-B55 pathway is essential for controlling mitotic substrate phosphorylation and mitotic entry. Here, we investigate the effect of the knockdown of the Gwl substrate, Ensa, in human cells. Unexpectedly, Ensa knockdown promotes a dramatic extension of S phase associated with a lowered density of replication forks. Notably, Ensa depletion results in a decrease of Treslin levels, a pivotal protein for the firing of replication origins. Accordingly, the extended S phase in Ensa-depleted cells is completely rescued by the overexpression of Treslin. Our data herein reveal a new mechanism by which normal cells regulate S-phase duration by controlling the ubiquitin-proteasome degradation of Treslin in a Gwl/Ensa-dependent pathway.

[1] Université de Montpellier, Centre de Recherche de Biologie Cellulaire de Montpellier, Equipe Labellisée 'Ligue Contre le Cancer', CNRS UMR 5237, 1919 Route de Mende, 34293 Montpellier cedex 5, France. [2] The Kinghorn Cancer Centre, Garvan Institute of Medical Research, Darlinghurst, NSW 2010 Australia. [3] St. Vincent's Clinical School, Faculty of Medicine, UNSW, Darlinghurst, NSW 2010 Australia. [4] Institute of Molecular Genetics, CNRS UMR 5535, University of Montpellier, 1919 Route de Mende, 34293 Montpellier, France. Correspondence and requests for materials should be addressed to T.L. (email: thierry.lorca@crbm.cnrs.fr) or to A.C. (email: anna.castro@crbm.cnrs.fr)

A precise spatiotemporal regulation of DNA replication is crucial for the maintenance of genomic integrity. DNA must be replicated once and only once during each cell cycle. Additional rounds of replication within a given cell cycle result in gene amplification, polyploidy and/or other kinds of genomic instability. Under-replication or late DNA replication can also cause genome instability, as for common fragile sites for example. Correct DNA duplication involves the strictly ordered assembly of various protein complexes onto thousands of genomic sites that will be destined to serve as replication origins[1,2]. The origin recognition complex (ORC) first binds the replication origins. This complex promotes the binding of Cdc6 and Cdt1, two proteins that will subsequently facilitate the binding of the MCM proteins to form the pre-replication complex (pre-RC).

Pre-RC formation process starts in late M phase and continues during early G1 when cyclin-dependent kinase (Cdk) activity is low. The subsequent initiation of DNA replication involves the activation of the MCM complex via the recruitment of the replication proteins Cdc45 and GINS that occurs at G1/S transition when interphase Cdk activity increases[3]. It is known that Cdks globally orchestrate transition at origin-bound complexes regulating licensing and initiation events to ensure that each origin is fired only once per cell cycle. During S, G2 and M phases origin licensing is prevented by high levels of Cdk activity that phosphorylate and inactivate multiple pre-RC components. One of these components, Cdt1, is inactivated during S phase by SCF-Skp2-dependent degradation as a consequence of Cdk-dependent phosphorylation[4, 5]. Another replication factor, Cdc6, is also phosphorylated by Cdk during DNA replication and this phosphorylation downregulates its licensing activity by promoting nuclear exclusion[6–8]. Finally, ORC1 phosphorylation by Cdk during S phase reduces its chromatin affinity[9] and permits its export to the cytoplasm preventing the formation of new pre-RC[10]. Unlike its negative effect on origin licensing, Cdk activity positively regulates origin firing at G1/S transition. In humans, Cdk phosphorylates Treslin, the orthologue of yeast *Sld3*, promoting its binding to TopBP1, the subsequent recruitment of Cdc45 to the pre-RC and the activation of the MCM helicase[11–14]. Sld3 has also been shown to bind MCM2-7 helicase and Dbf4-Cdc7 kinase. This binding stimulates DDK-dependent phosphorylation of MCM2 decreasing the affinity of this protein for MCM5 and probably facilitating in this way helicase ring opening[15, 16]. Recent data suggest that Cdk activity, through the phosphorylation of Treslin, controls S-phase length by regulating fork number[14].

Cdk activity plays a pivotal role not only in S phase but also in mitosis. Mitosis in most animal cells is promoted by the activation of cyclin B-Cdk1. Once activated, this kinase can phosphorylate a large number of substrates, thereby promoting major cellular reorganisation such as nuclear envelope breakdown, chromatin condensation and spindle formation. However, the phosphorylation state of cyclin B-Cdk1 substrates during mitosis is not exclusively dependent on the activity of this kinase but also on the activity of the phosphatase responsible of its dephosphorylation. Extensive work during the last 4 years has shown that mitotic entry is the result of the activation of the cyclin B-Cdk1 kinase and the inhibition of the phosphatase PP2A-B55. We, and others demonstrated that the activation of the Greatwall (Gwl) kinase at the G2/M transition via the phosphorylation of its substrates Arpp19 and Ensa, promotes the inactivation of PP2A-B55, the phosphatase responsible of the dephosphorylation of cyclin B-Cdk1 substrates[17–20]. This inactivation results in the stable phosphorylation of these substrates and correct entry and progression through mitosis. Here, we investigated the effect of the knockdown (KD) of the Gwl substrate, Ensa, in human cells. Our data demonstrate that Gwl/Ensa/PP2A-B55 axis regulates S phase by controlling the Cullin-dependent degradation of the pivotal replication factor Treslin.

## Results

**Ensa KD promotes a delay of cells in S phase.** Ensa has been firstly identified as one of the substrates of Gwl whose phosphorylation promotes the inhibition of PP2A-B55 and subsequent mitotic entry in Xenopus egg extracts. Little is known about the role of this protein in human cells. We sought to understand whether the role of this protein is conserved in humans. Toward this end, we carried out a small interfering RNA (siRNA) KD in HeLa cells. We used two siRNAs directed against two different sequences of the messenger RNA (mRNA) of this gene (siEnsa1 and siEnsa2). Both siRNAs efficiently depleted endogenous Ensa, with western blot analysis clearly showing the loss of a band corresponding to the molecular weight of Ensa in the targeting siRNAs, but not in the scramble siRNA (siSC) transfections (Fig. 1a). siEnsa1 transfection also caused a 90% decrease of the levels of an ectopically overexpressed green fluorescent protein (GFP)-hEnsa (Fig. 1b, c). Fluorescence-activated cell scanning (FACS) analysis of asynchronous HeLa cells 48 h after transfection with the two Ensa siRNAs revealed, unexpectedly, a dramatic accumulation in S phase, along with the apparition of a small subG1 population (Fig. 1d). A similar phenotype, although less striking, was observed when U2OS cells were knocked down of Ensa (Fig. 1e, f). This accumulation of cells in S phase was confirmed by bivariate FACS and indirect immunofluorescence after a brief BrdU incorporation and by EdU staining and immunofluorescence (Fig. 1g–i, respectively).

To further characterise this phenotype, we synchronised the cells in S phase by thymidine treatment for 24 h, 1 day after siRNA transfection. Since both Ensa siRNAs behaved similarly, we mostly used siEnsa1 for the rest of the study. Synchronised HeLa and U2OS cells were then analysed by FACS at different times after release from the thymidine arrest. Ensa knocked down cells remained in S phase as long as 10 h (HeLa) or 14 h (U2OS) after release, much longer than control cells, which already passed through mitosis and entered the next G1 by that time (Fig. 2a, b). To determine S-phase length, we performed a bromodeoxyuridine (BrdU)/5-ethyl-2′-deoxyuridine (EdU) double labelling in asynchronous HeLa cells treated with siSC or siEnsa1 RNA (Fig. 2c). Cells were pulse-labelled for 30 min with EdU, then washed, maintained in the medium before being pulsed again

**Fig. 1** Ensa KD strongly delays cells in S phase. **a** HeLa cells were treated with scramble siRNA (siSC) or Ensa siRNA#1 (siEnsa1) or #2 (siEnsa2) for 48 h. **a** Ensa levels and, as a loading control, β-tubulin(βTub) were checked in these cells by immunoblot. **b** Ensa-GFP was overexpressed in control cells (parental or transfected with siSC) and in cells knocked down of Ensa with siEnsa1 and GFP fluorescence intensity was analysed by FACS. **c** GFP levels in **b** were quantified by densitometry using ImageJ and represented as % of fluorescence towards GFP levels on parental cells. **d** HeLa cells, treated with SC siRNA (siSC) or siEnsa1 or siEnsa2, were stained with propidium iodide and analysed by FACS. DNA content (*x*-axis, propidium iodide) is represented towards the relative number of cells (*y*-axis, counts). *Coloured panel* represents a merge of all conditions. The quantification of % of total cells in SubG1, G1, S and G2/M phases in each cell type is represented as a *bar histogram*. **e** As for **d** except that U2OS instead of HeLa cells where used. **f** Endogenous Ensa levels in U2OS treated with siSC and siEnsa1 were analysed by western blot and shown. **g** Cells treated with control SC siRNA (*left*) or siEnsa1 (*right*) were incubated in the presence of BrdU for 30 min at 48 h post transfection and analysed by bivariate FACS. Intensity of BrdU incorporation and DNA content was analysed by using anti-BrdU antibodies and 7-AAD staining, respectively. *Boxed area* indicates the percentage of cells in S phase (incorporating BrdU), and in G1 (2n DNA content non incorporating BrdU) or G2/M (4n DNA content) phases (Flow Jo analysis). **h** Cells treated with siSC or siEnsa1 or siEnsa2 were incubated in presence of EdU for 60 min at 48 h post transfection and incorporated EdU was subsequently detected by Click-iT reagent. *Scale bar*, 5 μm. **i** The percentage of EdU-positive cells towards the total cell number (determined by differential interfere contrast microscopy (DIC)) in each condition was represented as a *bar graph* of the mean value ± standard deviation

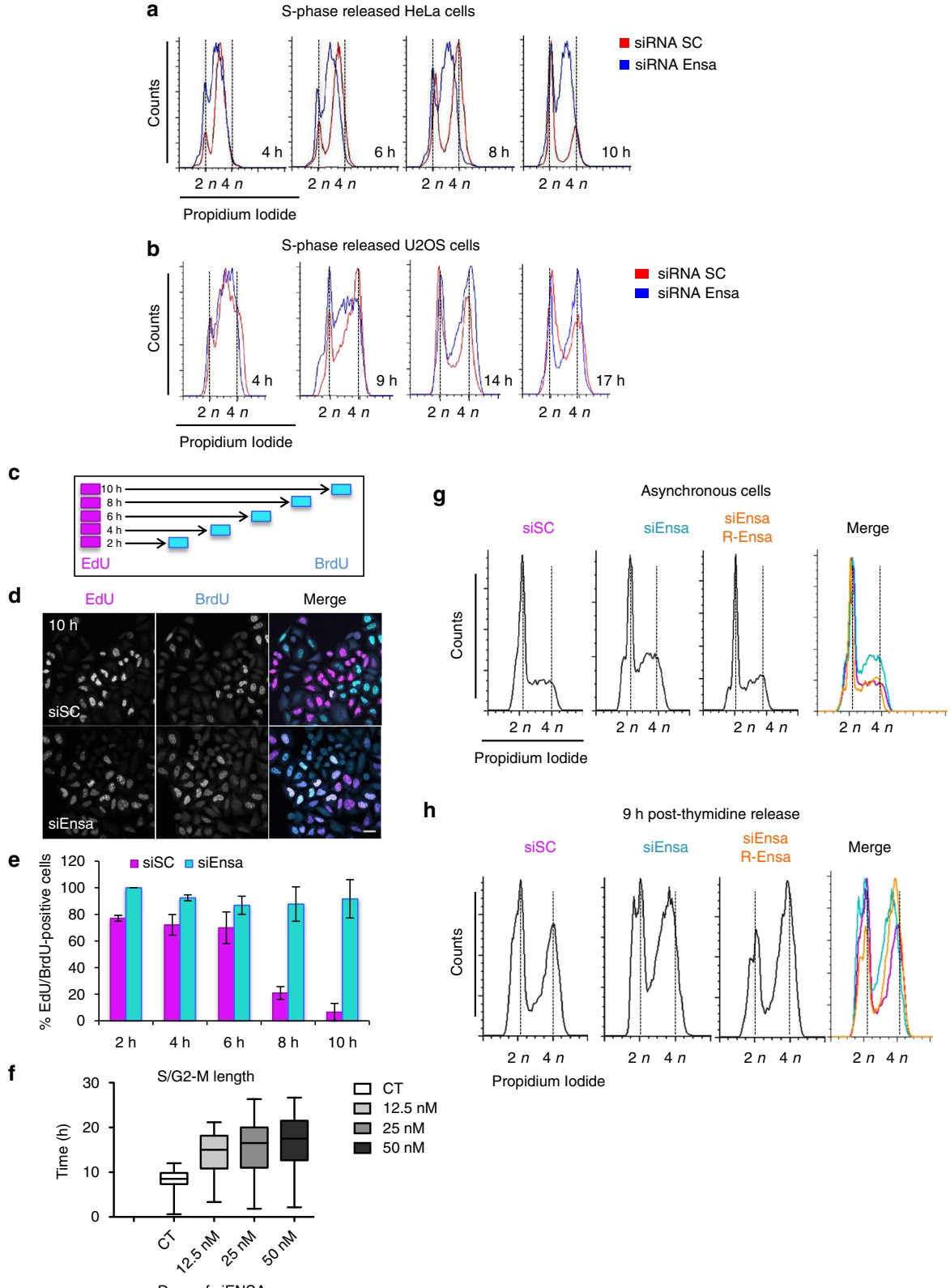

with BrdU (30 min) at 2, 4, 6, 8 or 10 h after EdU and finally fixed for immunofluorescence (Fig. 2d). S-phase length was calculated by measuring the percentage of BrdU/EdU-double-positive cells at each time point (Fig. 2e). This percentage decreased gradually in siSC cells reaching a minimum at 10 h after the first pulse, which suggests that S phase lasts around >10 h in these cells. In contrast, the percentage of double-positive siEnsa cells remained high throughout the experiment, indicating that S phase in these cells was much longer than 10 h.

To further examine the specificity of the phenotype induced by Ensa depletion, we investigated the effect of increasing doses of siEnsa on S–G2 length. HeLa cells were transfected with the

indicated doses of siRNA and synchronised by thymidine block. S-phase synchronised cells were then released and followed by time-lapse microscopy. The time between thymidine release and cell rounding (mitotic entry) was calculated as previously described[18]. Similar to FACS, siSC cells progressed through S phase and began entering mitosis on average 8.5 h after release. Increasing doses (12.5, 25 and 50 nM) of siEnsa showed a corresponding increase in S–G2 length with median values of 15, 16.5 and 17.5 h, respectively (Fig. 2f). Finally, to investigate the specificity of our siEnsa, we generated HeLa cells that stably expressed a siRNA-resistant transgene of human Ensa (R-Ensa). Overexpression of R-Ensa had no noticeable effect on S-phase progression (Supplementary Fig. 1). HeLa and R-Ensa-HeLa cells were transfected with siRNA (scramble or siEnsa) for 24 h, then later synchronised (Fig. 2h) or not (Fig. 2g) for 21 h with thymidine and analysed by FACS at 9 h after release. As expected, control HeLa cells treated with siEnsa displayed a dramatic delay in S phase in both asynchronous cells and in synchronic cells 9 h after release. In contrast, at this time, the majority of scramble siRNA-treated cells had completed S phase and were undergoing mitosis. Importantly, stable overexpression of R-Ensa in HeLa cells rescued the S-phase delay induced by the KD of Ensa. These data indicate that our siRNAs specifically induce Ensa KD and that this depletion dramatically extends S phase.

**Ensa KD extends S phase by decreasing fork number.** The increase of S-phase length induced by Ensa KD could be the result of a decrease of fork rate or fork number. To discriminate between these hypotheses, we used DNA combing. Newly replicated DNA was labelled by sequentially treating cells for 15 min with two different thymidine analogues: iododeoxyuridine (IdU), then chlorodeoxyurindine (CldU). Tracts of ongoing DNA replication were identified as adjacent IdU–CldU signals on individual DNA fibres (Fig. 3a, b). Mean fork rate was calculated by dividing tract length by pulse duration on each in an ongoing replication tract, whereas fork density was calculated by dividing the total number of bicolour forks by the total length of DNA fibres analysed after normalisation for the fraction of S-phase cells. No differences in fork rate was observed between siSC and siEnsa-treated cells (Fig. 3c), however, fork density significantly decreased in Ensa knocked down cells (median values: 1.50 forks per Mb in siSC vs 0.92 forks per Mb in siEnsa cells; Fig. 3d). This strongly suggests that increased DNA replication time in the Ensa-depleted cells was the result of a decreased number of active replication origins.

The Gwl/Ensa pathway is essential for mitotic entry and controls mitotic progression. During late mitosis and early G1, pre-RC formation occurs as a sequential assembly of the licensing factors Cdc6 and Cdt1 onto ORC-bound chromatin, which then recruit the MCM2-7 complex[2, 21]. It is possible that defects in mitotic progression due to Ensa depletion could perturb pre-RC formation and licensing during late M/early G1 phase resulting in a decreased number of replication origins and subsequent decreased fork number. In order to test this hypothesis, we designed an experiment in which synchronised cells were allowed to traverse mitosis and G1, thus making preRCs, before being transfected with siEnsa (4 h after nocodazole shakeoff) and 1 h later blocked in S phase by thymidine addition for 18 h to allow Ensa depletion (Fig. 3e). In this scheme, Ensa is depleted at a stage after origin licensing, thus any S-phase delay after release would indicate a role of Ensa in origin firing, not licensing. This was indeed the case, with siEnsa cells presenting the same slow S-phase progression as asynchronous siEnsa-transfected cells (Fig. 3f). To deeper analyse whether Ensa KD could affect origin licensing, we checked the levels of MCM proteins in the chromatin fraction of siSC and siEnsa HeLa-and U2OS-treated cells (Fig. 3g, h). Cytosolic (S2) and nuclear soluble (S3) fractions as well as chromatin fraction (P3) were purified from nuclei of these cells as described in Methods section and used for western blot analysis. As expected from the data above, the levels of the different chromatin-bound MCM did not significantly change in Ensa KD cells confirming that the depletion of this protein is not affecting origin licensing. We conclude that the decrease of fork density in Ensa-depleted cells is unlikely to be due to a defect in pre-RC formation, but rather in the firing of origins.

**Ensa KD does not alter spatiotemporal replication pattern.** Our data show that Ensa KD causes an extension of S phase. To get more insight into this phenotype, we analysed the spatiotemporal pattern of replication foci in Ensa-depleted HeLa cells stably expressing a proliferating cell nuclear antigen (PCNA)-targeting chromobody fused to GFP, allowing the visualisation of endogenous PCNA foci[22]. After siRNA transfection, cells were imaged for 72 h by time-lapse microscopy. The direct visualisation of replication factories in vivo confirmed that S-phase length was dramatically increased in Ensa knocked down cells with a mean time corresponding to 7 h in scramble and 11 h in siEnsa-treated cells (Fig. 4a, Supplementary Movies 1–6).

We could distinguish early, mid and late replication factories in both scramble and siEnsa-treated cells following previous described nomenclature as shown in Fig. 4b[23]. However, interestingly, although we could not quantify differences in intensity or size due to variable basal GFP intensities and different replication factory shapes, we noticed clear alterations on the localisation and the size of mid-replication factories between scramble and siEnsa-treated cells. Specifically, in scramble-treated cells, mid-replication factories were small, creating a continuous and complete circumference around nucleoli. In contrast, in

**Fig. 2** Ensa KD increases S-phase length. **a** HeLa cells treated with SC or Ensa siRNA were blocked in G1/S by thymidine and released in fresh medium. Samples were taken at 4, 6, 8 and 10 h after release and processed by FACS to follow S-phase progression. **b** As for **a** except that U2OS cells were used and released at 4, 9, 14 and 17 h. **c** Scheme representing the experiment in which HeLa cells were treated with SC or Ensa siRNA for 48 h, pulsed with 10 μM EdU for 30 min and, after washing, incubated in fresh medium containing 5 μM thymidine for 2, 4, 6, 8 and 10 h. After a second pulse of 30 min with 10 μM, BrdU cells were fixed for immunofluorescence. **d** EdU/BrdU double staining of siSC and siEnsa-treated cells at 10 h after thymidine release is shown. EdU-positive cells are labelled in *magenta* and BrdU-positive cells in *cyan*. Resulting white co-localisation are cells still in S phase. *Scale bar*, 20 μm. **e** The % of EdU/BrdU-positive cells in **d** was counted and represented as a *bar graph* of the mean value ± standard deviation. A minimum of 80 cells was counted in each time point. **f** HeLa cells were or not (CT) transfected with increasing doses of siEnsa (12.5, 25 and 50 nM) and 24 h later synchronised in S phase by thymidine block. Cells were then released and followed by time-lapse microscopy. Time after thymidine release and cell rounding (G2/M transition) for each cell at each siRNA concentration was recorded and represented as a *box-and-whiskers* diagram showing median and minimum to maximum values. A minimum of 50 cells was counted per condition. **g** Asynchronous control or HeLa cells stably expressing a siEnsa-resistant transgene of human Ensa (R-Ensa) were transfected with siEnsa1 or with SC siRNA (only for parental cells). Cells were released 24 h later, recovered and used for FACS analysis. **h** As for **g**, except that cells were blocked in G1–S 24 h after transfection by thymidine and 9 h after release used for FACs analysis

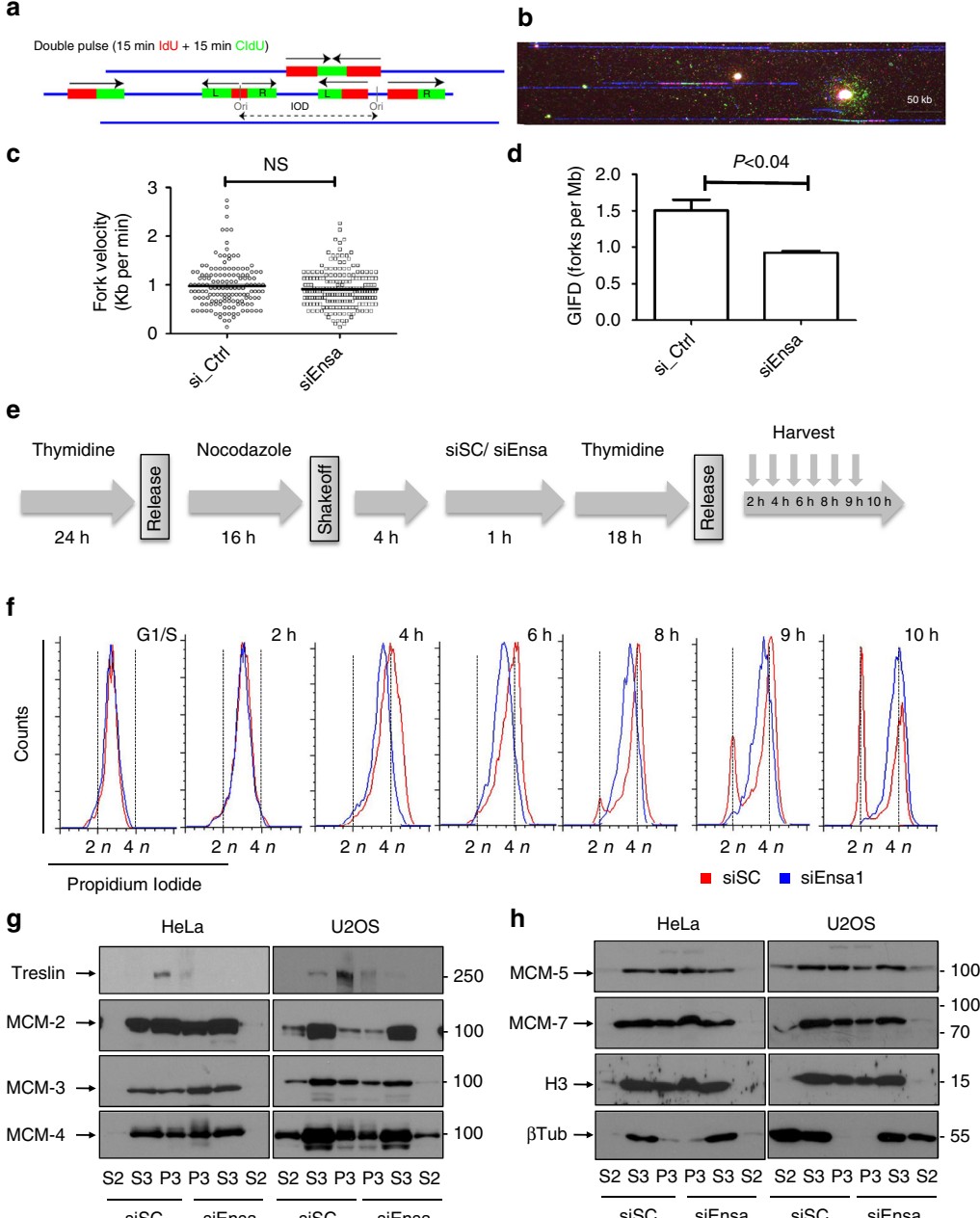

**Fig. 3** Ensa KD increases S-phase length by decreasing fork number without affecting origin licensing. **a** Diagram depicting IdU (*red*)–CldU (*green*) double pulse analysis by DNA combing. **b** Representative image of a triple-stained DNA fibre: IdU (*red*)–CldU (*green*)–DNA (*blue*). **c** Fork velocity (Kb/min) and **d** global instant fork density (GIFD) values (fork per Mb) are represented. Mean ± standard deviation and *p*-value obtained by using two-tailed unpaired Student's *t*-tests, are represented. Data pooled from four biological replicates. **e** Diagram representing the experiment in which HeLa cells were blocked in G1–S phase by thymidine for 24 h and subsequently released with fresh medium containing nocodazole (50 ng/ml) for a supplementary period of 16 h before shakeoff. Four hours later, released cells were transfected with siSC or siEnsa, and after 1 h, supplemented with fresh medium containing thymidine for an additional 18 h incubation. Cells were then released and recovered at the indicated times for FACS analysis. **f** FACS profiles of SC (*red*) and Ensa (*blue*) siRNA-treated cells at different time points after release are shown. **g** HeLa and U2OS cells were transfected with scramble (siSC) or Ensa (siEnsa) siRNAs and 48 h later processed for chromatin isolation as reported in Methods section. Cytosolic (S2) and nuclear (S3) soluble fractions as well as chromatin fraction (P3) were then used for western blot analysis to determine Treslin and MCM 2, 3 and 4 levels. **h** As for **g**, except that MCM 5 and 7, histone H3 and β-tubulin levels were checked. IOD, inter-origin distances; Ori, replication origin

siEnsa-treated cells, these factories were dramatically bigger. Moreover, the nucleoli were hardly visualised in siEnsa-treated cells, with PCNA staining no longer continuously surrounding nucleoli, but rather displaying discrete, pointed localisation (Fig. 4c, Supplementary Movies 1–6).

We next measured the time after thymidine release when each type of factory was visualised. Our quantification data

demonstrate that the general spatiotemporal patterns of replication foci were not perturbed with all (early, mid and late S phases) patterns being extended proportionally (Fig. 4d, Table 1, Supplementary Fig. 2a and Supplementary Movies 1–6). Similar results were obtained when the time between early–mid–late phase transitions were measured by using double EdU/BrdU staining and immunofluorescence (Supplementary Fig. 2b, c).

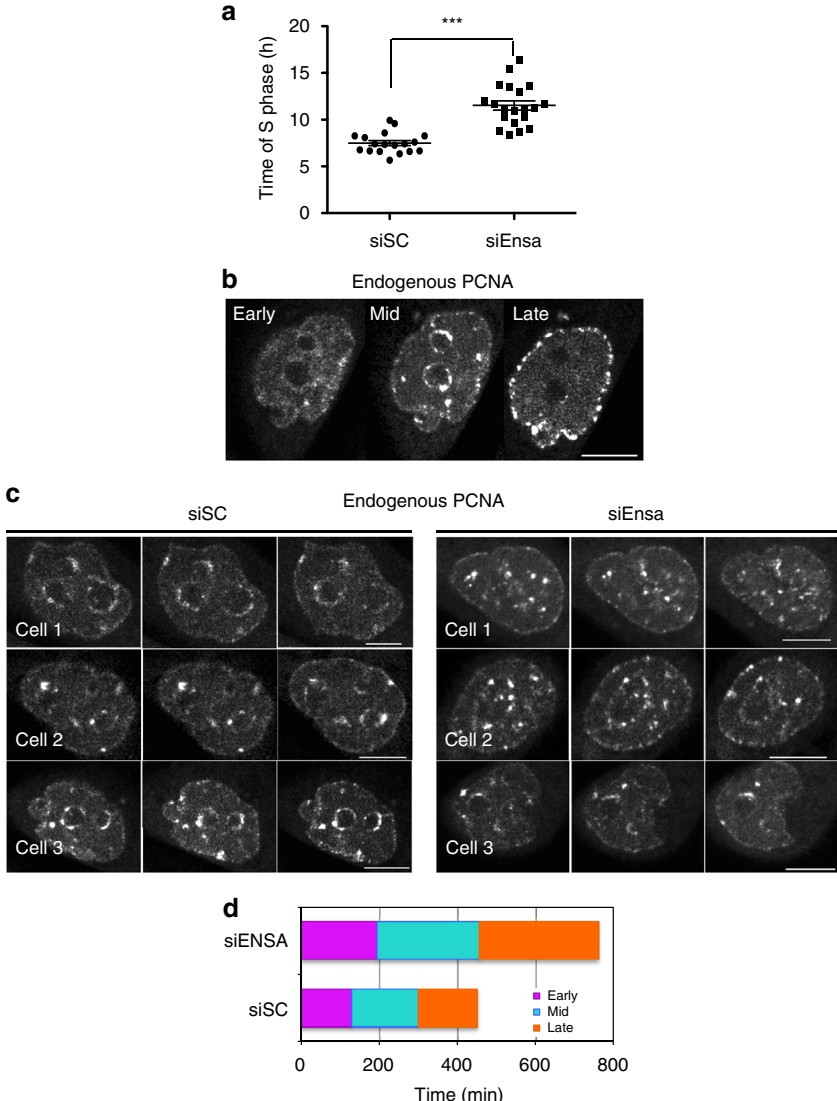

**Fig. 4** Depletion of Ensa extends S phase without significantly altering the overall spatiotemporal patterns of replication. **a** HeLa chromobody cells were transfected with SC and Ensa siRNA and imaged by time-lapse microscopy for 72 h and S-phase duration visualised by GFP-nanobody-dependent staining of PCNA, recorded and represented as scatter dot blot ($n = 18$ for siSC and $n = 20$ for siEnsa). Two-tailed unpaired Student's $t$-tests was performed to determine the statistical relevance ***$p < 0.0001$. Data pooled from three biological replicates. **b** Representative images of early, mid and late replication factories from HeLa chromobody control cells used for time-lapse microscopy. *Scale bar*, 10 μm. **c** Representative images of mid-replication origins from Supplementary Movies 1–6. Images corresponding to three Z-planes of three scramble and three siEnsa-treated HeLa cells stably expressing a chromobody directed to PCNA. Plans correspond to 4 h, 3 h 20 min and 4 h 15 min time points after DNA replication started in siSC-treated cells 1, 2 and 3 and to 5 h 15 min, 5 h 15 min and 6 h 50 min time points in siEnsa-treated cells 1, 2 and 3 from movies on supplementary data. *Scale bar*, 10 μm. **d** HeLa chromobody cells treated as in **c** were imaged by time-lapse microscopy and the length of the early, mid and late S-phase patterns was recorded and represented as a *bar graph*

These data indicate that S-phase delay is likely due to a general inhibition of the firing of all, early, mid and late origins, suggesting that this delay could stem from a general defect in DNA replication initiation factor(s).

**The knockout of Gwl also promotes partial S-phase extension**. We next checked the effect of the depletion of Gwl on S-phase length. To do this, we used Gwl(Lox/Lox) or control mouse embryonic fibroblasts (MEFs)[17] in which Gwl ablation is induced by the transduction of GFP-Cre-expressing adenoviruses (Fig. 5a). We did not observe any effect when control MEFs were infected with the plasmid coding for GFP-Cre (Fig. 5b) or when Gwl(Lox/Lox) MEFs were infected with the GFP-expressing adenoviruses (Fig. 5c). As shown previously, the acute expression of GFP-Cre in asynchronous Gwl(Lox/Lox) MEFs promoted an accumulation of mitotic cells and cells with 8n DNA content due to the lack of chromosome segregation and/or to cytokinesis defects[17, 18] (Fig. 5d). However, we also observed a partial accumulation of cells in S phase. To further investigate this phenotype, MEF Gwl(Lox/Lox) cells were serum-starved for 3 days and subsequently infected with GFP as a control (Fig. 5e) or GFP-Cre-expressing adenoviruses (Fig. 5f). Five hours later, cells were supplemented with fresh medium containing aphidicolin for 24 h and were recovered for FACS analysis at 5 and 14 h after release. As expected, both control and Gwl-ablated MEFs were in S phase at 5 h after release, however, at 14 h, control cells were out of S phase, whereas Gwl knockout cells significantly accumulated at this phase of the cell cycle. To quantify this S phase accumulation, we synchronised infected control (GFP) and knockout (GFP-Cre) MEFs by the addition of aphidicolin for 24 h

**Table 1 Spatiotemporal patterns of replication of Ensa KD cells**

| | Early | | Mid | | Late | |
|---|---|---|---|---|---|---|
| | siSC | siEnsa | siSC | siEnsa | siSC | siEnsa |
| Mean time (min) | 128.5 | 192.75 | 170 | 261 | 151.6 | 305 |
| Standard deviation | 40.7 | 80.2 | 40.8 | 100.9 | 67.01 | 89.8 |
| n (cells) | 18 | 20 | 18 | 20 | 18 | 20 |
| p-value (t-test) | | 0.0039 | | 0.0012 | | 0.0021 |

Length of the early, mid and late S-phase patterns were counted in cells from Fig. 4. Mean, standard deviation, number of cells and p-value obtained by using two-tailed unpaired Student's t-tests between siSC and siEnsa treatments are represented in the table. Data correspond to three different experiments

and we performed a pulse of 30 min of BrdU at different times post release. Cells were then fixed and immunostained with anti-BrdU antibodies. The number of S-phase cells in control and Gwl knockout cells were then counted at each time point and compared. As shown in Fig. 5g, in control MEFs, BrdU incorporation peaked at 6 h, decreasing slightly till 10 h before significantly decreasing from 13 to 18 h, indicating that S-phase length was ~10 h. In contrast, in Gwl knockout MEFs, high BrdU staining was maintained until 15 h after release and did not start to decrease until 18 h, suggesting that S phase was also partially extended by Gwl knockout in these cells.

**Active Gwl and Ensa are localised in the nucleus in S phase.** The above results suggest that Gwl, likely by phosphorylating Ensa, participates to the control of S phase. We therefore checked the localisation of both Ensa and Gwl during S phase. Previous reports[17, 18] showed that Gwl was localised in both nuclei and cytoplasm, however, whether this nuclear localisation is observed during S phase or the other phases of the cell cycle is not known. We checked nuclear localisation of Gwl by immunofluorescence in BrdU-labelled cells. As shown in Fig. 6a, nuclear localisation of Gwl was constant throughout interphase and no differences were observed when BrdU-positive and -negative cells were compared. In contrast, nuclear localisation of Ensa was significantly increased in S-phase cells suggesting that Ensa is relocalised into the nucleus during DNA replication.

Nuclear localisation of Ensa in S phase is compatible with a role of this protein in DNA replication. We thus further investigated if Ensa was phosphorylated and activated during S phase by Gwl. To that, we checked Ensa phosphorylation in Ensa immunoprecipitates of HeLa cells during the different phases of the cell cycle by using a phospho-antibody that recognises the conserved phosphorylated site of Gwl in both human Arpp19 (S62) and human Ensa (S67). Phosphorylation of Ensa by Gwl significantly increased in S phase compared to G1/S but to a lower extent than in mitosis (Fig. 6b). We next checked the levels of active Gwl on the chromatin during S phase. Hence, thymidine-blocked HeLa cells were lysed to recover cytoplasm and nuclei at 2, 4, 6 and 8 h after release. Chromatin-associated proteins were then extracted by using a buffer containing 300 mM (N3) or 600 mM of NaCl (N6). As shown in Fig. 6c, Gwl was present and bound to the chromatin throughout S phase although to a lesser extent than in cytoplasmic fraction. Moreover, phosphorylation and activation of Gwl was observed in nuclear but not in cytoplasmic fractions of thymidine-blocked cells. Gwl phosphorylation in nuclear fractions was maintained 2 and 4 h after release, decreased at 6 h and disappeared at 8 h when S phase is completed. These results indicate that Gwl and Ensa are present and active on the chromatin during S phase.

**Ensa KD lowers Treslin protein levels.** It is known that the Gwl/Ensa pathway regulates mitotic entry and progression by promoting the inhibition of PP2A-B55, the phosphatase responsible of cyclin-Cdk substrate dephosphoryaltion[19, 24]. We investigated the effect of Ensa KD on the global phosphorylation of Cdk substrates by using an antibody directed against the phosphorylated serine in the Cdk consensus motif. As shown in Fig. 7a, b, we observed a significant decrease of the phosphorylation of cyclin-Cdk substrates in Ensa KD cells. We hypothesised that the Gwl/Ensa pathway might be activated in S phase to permit the phosphorylation of key Cdk substrates responsible for the correct unfolding of S phase. One major Cdk substrate whose phosphorylation controls S-phase length is the TopBP1-interacting protein, Treslin[14]. We therefore checked whether Treslin was dephosphorylated upon Ensa KD. To our surprise, the main phenotype that we observed after siEnsa treatment was a dramatic drop of this protein in both HeLa and in U2OS cells (Fig. c and d) and in U2OS cells overexpressing Treslin (Fig. 7e, f). This decrease was not due to a decline of the mRNA transcripts of this protein (Fig. 7g). We observed a similar drop of Treslin levels in Gwl knockout MEF cells supporting the hypothesis that the role of Ensa in S phase is mediated by the Gwl-dependent phosphorylation of this protein (Fig. 7h, i). Since no data about a putative control of Treslin amount during cell division were reported, we checked endogenous Treslin levels throughout the cell cycle. Our data show that the endogenous levels of this protein are high during M/G1, drop during S phase and increase again in G2 to reach maximum levels at M phase suggesting that Treslin levels would be tightly controlled during DNA replication (Supplementary Fig. 3). Interestingly, although endogenous Treslin was high during mitosis, a decrease of this protein after Ensa KD was also observed during mitosis indicating that the Ensa-dependent Treslin stabilisation pathway is also active during this phase of the cell cycle (Fig. 7j, k).

**Treslin rescues S-phase length in Ensa KD cells.** The above data show that Ensa KD induces a dramatic decrease of Treslin levels. Lower Treslin levels might decrease the number of fired replication origins, and consequently lead to a longer S phase. If this was the case, recovering normal Treslin levels should rescue the Ensa KD phenotype. We therefore tested whether Treslin overexpression restores S-phase length in Ensa knocked down cells. Ensa was knocked down by siRNA in a U2OS cell line stably overexpressing Treslin (Fig. 8a), and Treslin levels (Fig. 8b, c) along with S-phase duration was measured by FACS at 7.5 and 10 h post-thymidine release (Fig. 8d, e, respectively). As shown in Fig. 8a, Ensa was efficiently knocked down in both parental and Treslin-overexpressing UO2S cells. The Treslin-U2OS stable cell line (CT) expressed approximately four times more Treslin than scramble treated U2OS cells (compare CT in Treslin-U2OS cells with sSC in U2OS cells in Fig. 8c). As expected, we saw a decrease of Treslin amounts in U2OS control cells after Ensa KD (compare siSC vs siE1 and siE2 in U2OS cells in Fig. 8b, c). Treslin levels also decreased in the Treslin-U2OS stable cell line depleted of Ensa, however the levels of this protein remained above endogenous siSC-treated controls (Fig. 8c, compare siSC vs siE1 and siE2 in Treslin-U2OS cells). Importantly, FACS data analyses showed that S-phase progression is as fast after Ensa KD in U2OS cells overexpressing Treslin than in normal U2OS cells treated with scramble siRNA. In contrast, in U2OS cells treated with siEnsa1 or 2, S phase was extended as before. These data support the idea that the S-phase extension caused by Ensa KD stems from the decrease of Treslin protein levels, which is one of the limiting factors regulating the firing of replication origins and controlling replication timing.

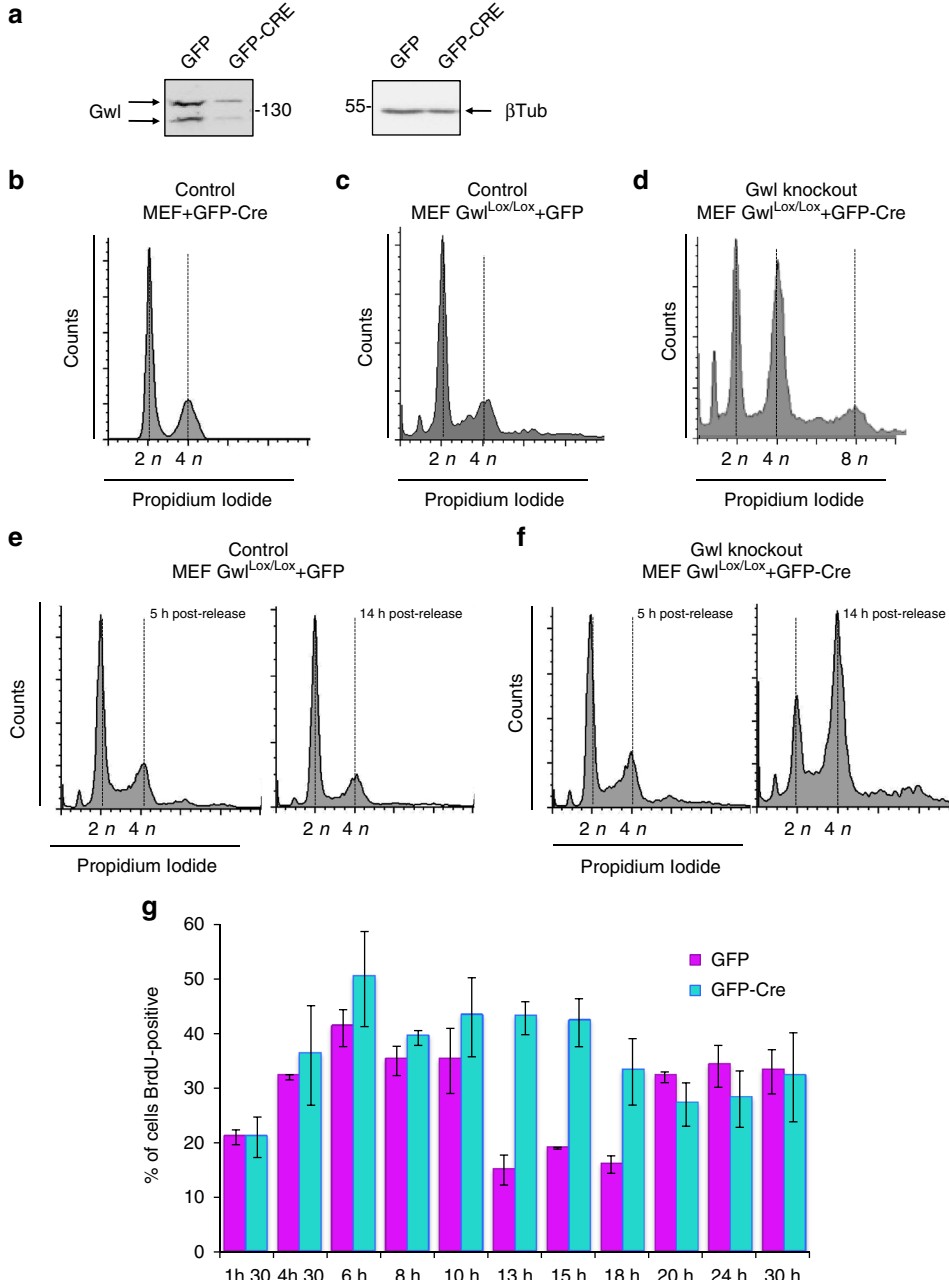

**Fig. 5** The knockout of Greatwall, the upstream kinase of Ensa, also results in partial S-phase accumulation. **a** Control or MEF Gwl(Lox/Lox) cells were serum-starved for 3 days, infected for 5 h with GFP or GFP-Cre-expressing adenoviruses as indicated, stimulated to re-enter the cell cycle with fresh serum-supplemented media and recovered for western blot with the indicated antibodies. **b** FACS analysis 48 h after infection in control MEFs infected with GFP-Cre adenoviruses. **c** Gwl$^{Lox/Lox}$ MEFs infected with control GFP adenovirus. **d** Gwl$^{Lox/Lox}$ MEFs infected with GFP-Cre adenovirus. **e** MEF Gwl$^{Lox/Lox}$ cells were serum-starved for 3 days and subsequently infected with GFP-expressing adenovirus. Five hours later, cells were supplemented with fresh medium containing aphidicolin for 24 h. S-phase synchronised cells were then released and recovered at 5 and 14 h for FACS analysis. **f** As for **e**, except that GFP-Cre-expressing adenoviruses was used. **g** S-phase synchronised cells infected with GFP (control) and GFP-Cre (Gwl knockout) adenoviruses as in **e** and **f** were released and a pulse of 30 min of BrdU at the indicated time points was performed. Cells were then fixed and immunostained with anti-BrdU antibodies. The number of S-phase cells in control and Gwl knockout cells were counted at each time point, compared and represented as a *bar graph* of the mean value of three different samples ± standard deviation. A cell number between 175 and 565 were counted for each time point

**Ensa stabilises Treslin by preventing its dephosphorylation.** The findings above suggest that Ensa depletion either reduces Treslin translation or promotes Treslin degradation. No data about a putative regulation of Treslin protein levels have ever been reported. Thus, we used the proteasome inhibitor MG132 to check whether Treslin is degraded by the proteasome. No major

effects in cell cycle progression were observed after MG132 addition in HeLa cells (Supplementary Fig. 4a). As previously reported[25], the addition of this proteasome inhibitor promoted a small increase of S-phase cells that was identical in all parental, siSC and siEnsa-treated U2OS cells and that in the latter, accumulated with the S-phase delay already present in non-treated

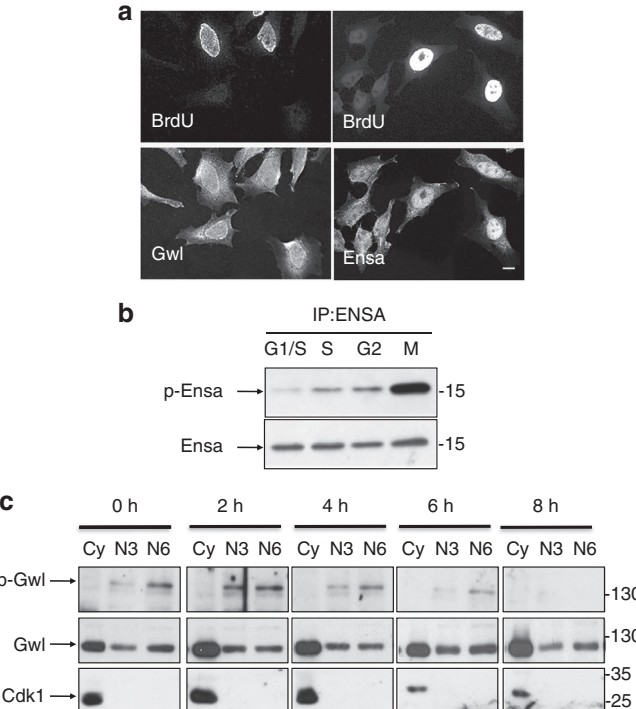

**Fig. 6** Gwl and Ensa are active and localised into the nucleus during S phase. **a** Asynchronous HeLa cells were incubated with BrdU for 30 min and subsequently fixed and used for immunofluorescence with anti-BrdU and anti-Gwl or anti-Ensa antibodies. *Scale bar*, 10 μm. **b** Ensa protein was immunoprecipitated in G1/S, S, G2 or M phase synchronised HeLa cells and used for western blot using antibodies recognising Ensa (Ensa) protein or the phosphosite S67 of this protein (p-Ensa). **c** HeLa cells were synchronised in S phase by 24 h thymidine incubation and at the indicated times after release, cells were lysed with hypotonic buffer and the cytoplasmic fraction was recovered (Cy). Nuclear pellet was split into two equal parts and resuspended with NETN buffer containing either 300 mM or 600 mM NaCl. After 30 min incubation, samples were centrifuged and supernatants (N3 for 300 mM or N6 for 600 mM) were used for western blots with Cdk1 and anti-phospho Gwl S875 antibodies. Cdk1 levels were used to confirm the purity of cytosolic and nuclear fractions

Ensa KD cells (Supplementary Fig. 4b). Treslin significantly accumulated after MG132 addition in both HeLa (Fig. 9a) and U2OS cells (Fig. 9b). A comparable accumulation was observed in Ensa knocked down cells although, as expected, in this case Treslin basal levels were significantly lower than in control cells. Treslin levels also increased when the activity of the Cullin-dependent E3 ubiquitin ligases were blocked in both scramble and Ensa siRNA-treated cells by using the NEDDylation activating enzyme inhibitor, MLN4924 (Fig. 9a, b). These results indicate that Treslin is an unstable protein that is likely degraded by Cullin-proteasome pathway. Since Treslin accumulated in both control and Ensa knocked down cells, these data also suggest that Ensa depletion does not prevent Treslin translation but rather promotes a drop of Treslin levels by increasing the degradation of this protein. Accordingly, Treslin levels also dramatically decreased in U2OS cells stably overexpressing Treslin (Treslin-U2OS) when treated with siEnsa. Moreover, the addition of MG132 to these cells promoted the accumulation of Treslin (Fig. 9c).

The binding of certain Cullins, notably of the Cul1-CRL complex (SCF) to their substrates is mediated by phosphorylation. It is known that Treslin is phosphorylated by Cdk at S phase.

We thus tested whether Cdk-dependent phosphorylation of Treslin could regulate the proteolysis of this protein. To that, we checked the degradation pattern of ectopic wild type or of a phosphomimetic Treslin mutant on the T968 and S1000 Cdk sites (TESE mutant) in HeLa and U2OS cells (Fig. 9d, e, respectively). These cells were supplemented with cycloheximide to prevent vector promoter-induced continuous synthesis of the ectopic Treslin. No effect of cycloheximide addition in cell cycle progression was observed (Supplementary Fig. 5). As expected, wild-type Treslin levels drop in both cell lines when protein synthesis was inhibited by the addition of cycloheximide. On the contrary, the phosphomimetic TESE mutant was stable throughout the experiment under these conditions suggesting that the phosphorylation of Treslin by Cdk stabilises this protein. Similar results were obtained in siEnsa-treated cells (Fig. 9f). Since the Gwl/Ensa pathway controls the phosphorylation of Cdk substrates by inhibiting PP2A, we next tested whether Ensa KD could promote Treslin degradation by inducing the PP2A-dependent dephosphorylation of this protein. HeLa and U2OS cells were transfected with scramble or Ensa siRNAs and 24 h later, supplemented with a low dose of the PP2A inhibitor okadaic acid (5 nM) for a supplementary 24 h incubation. The cellular levels of Treslin were then measured by western blot (Fig. 9g, h, respectively) and cell cycle distribution in treated HeLa cells analysed by FACs (Fig. 9i). Scramble-treated cells displayed a normal FACs profile and identical Treslin levels indistinctly of the presence or absence of okadaic acid. Conversely, as expected, Ensa knocked down cells accumulated in S phase and displayed a drop of Treslin in the absence of okadaic acid, a phenotype that was significantly rescued when the PP2A inhibitor was supplemented. These data suggest that Ensa, via PP2A inhibition, controls Treslin stability by modulating Cdk-dependent phosphorylation of this protein.

## Discussion

The Gwl/Arpp19-Ensa/PP2A-B55 axis was originally identified in Xenopus egg extracts[26–28] and subsequently reported in other models such as Drosophila[29] and human cells[17, 18]. Activation of the Gwl kinase induces the phosphorylation of Ensa, promoting their binding to and inhibition of the phosphatase PP2A-B55[19, 20]. Inhibition of PP2A-B55 results in the stable phosphorylation of cyclin B-Cdk1 substrates and mitotic entry.

So far, the activation of this pathway has been reported in mitotic cells, however, it has been recently demonstrated that Gwl has additional roles and is involved in regulating Akt activation pathway[30]. This raises the possibility that the Gwl-Ensa-PP2A pathway is functional outside of mitosis. To test this, we investigated the role of Ensa in human cells using siRNA KD. Unexpectedly, the first phenotype that was observed in knocked down cells was a dramatic extension of S phase. This extension was displayed in both HeLa and U2OS cells and was specific to Ensa depletion, since the same siRNA treatment in a cell line stably expressing a siRNA-resistant Ensa transgene completely rescued the phenotype. Interestingly, S-phase delay induced by Ensa KD was mediated by a decrease of the number of fired replication origins without affecting either origin licensing or fork velocity. In addition, the 'in vivo' analysis of replication factory dynamics in Ensa KD cells using a anti-PCNA GFP chromobody-expressing cell line revealed that the general spatiotemporal pattern of early, mid and late replication factories was not altered during S phase, despite a clear extension of the time each of these patterns lasted. Together, these data indicate that the spatiotemporal extension is likely induced by a general decrease of the firing of early, mid and late replication origins resulting in an increase of S-phase duration.

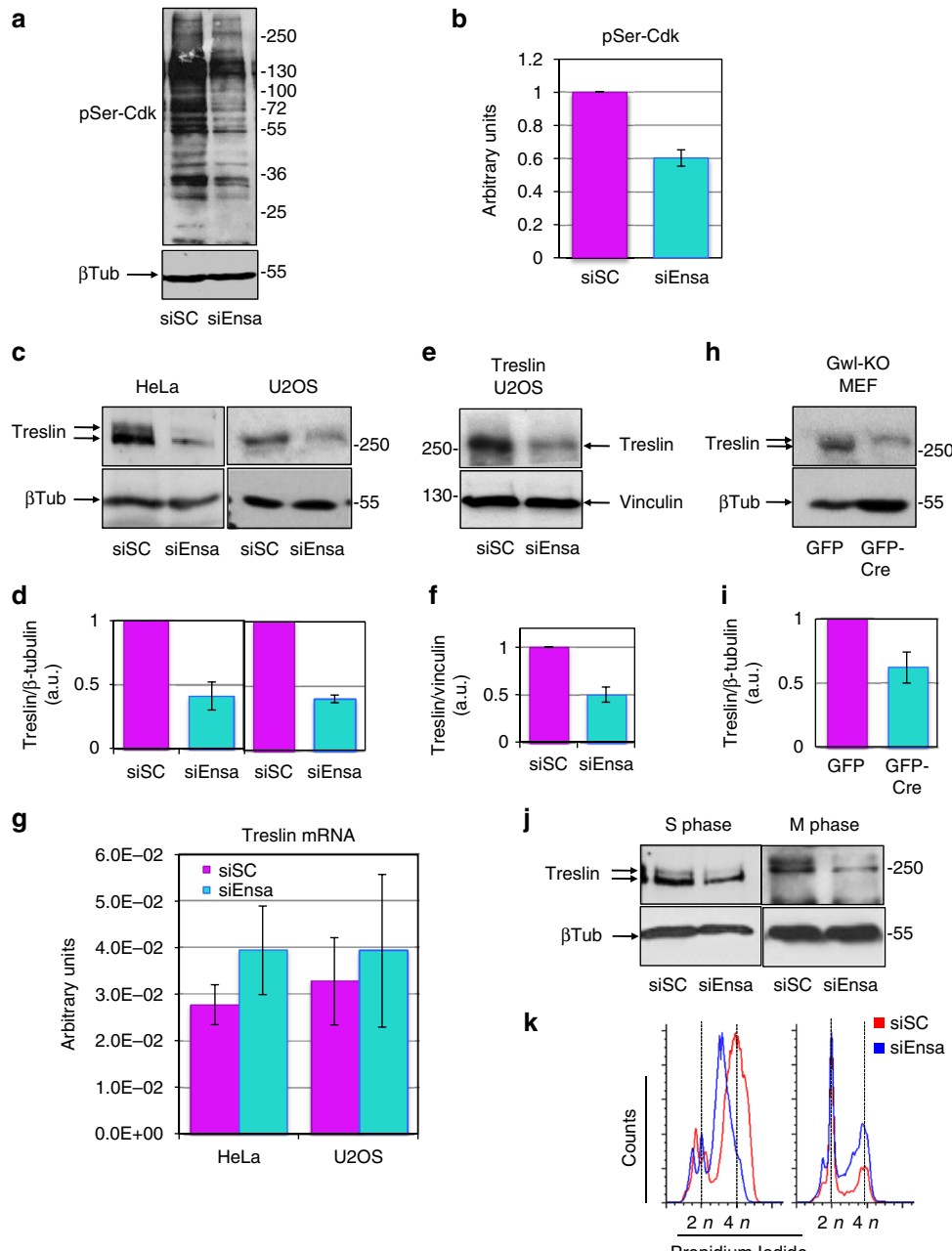

**Fig. 7** Depletion of Ensa decreases Treslin protein levels. **a** Representatitve immunoblotting of phosphorylated Cdk substrates in control (siSC) and Ensa siRNA-treated cells (siEnsa). **b** Phospho-Ser Cdk intensities on the western blot were measured by densitometry using ImageJ software in five independent experiments. Mean intensity was normalised to controls and represented as *bars* and *standard deviation*. **c** Immunoblotting of Treslin and, as loading controls, β-tubulin in HeLa and U2OS cells treated with scramble (siSC) or Ensa siRNA. **d** Mean Treslin/β-tubulin ratios were normalised to controls and represented as a *bar graph* ± standard deviation. **e** U2OS cells overexpressing Treslin were treated with scramble (siSC) or Ensa siRNA and used for western blot. **f** Mean Treslin/Vinculin ratios were normalised to controls and represented as a *bar graph* ± standard deviation. **g** Treslin mRNA levels were measured using real-time quantitative PCR in HeLa and U2OS treated with siSC or siEnsa1. Normalisation was made with the expression of the housekeeping gene *hsMRPL19*. Data are means ± standard error of mean. **h** Gwl[Lox/Lox] MEFs were transduced with GFP or GFP-Cre expressing adenoviruses and used for western blot 48 h post infection. **i** Mean Treslin/β-tubulin ratios were normalised to controls and represented as a *bar graph* ± standard deviation. **j** HeLa cells were transfected with (siSC) and Ensa (siEnsa) siRNAs, synchronised in S and M phases as indicated in Methods section and used for western blot. **k** Cell cycle distribution in cells from **j** was determined by FACS. All the results correspond to at least three different experiments

Interestingly, our data also reveal that this general decrease of early, mid and late origin firing in Ensa knocked down cells is the result of a dramatic drop of Treslin levels, which is an essential and limiting factor for origin firing[12, 14, 31]. Consistently, we show that the S-phase extension is suppressed when the original Treslin levels are recovered by ectopic expression despite Ensa KD.

Another key result of our study highlights the fact that Treslin is a highly unstable protein whose levels are likely regulated by Cullin-dependent ubiquitination and proteasome degradation. In addition, our data suggest that depletion of Ensa promotes an increase of Treslin ubiquitin/proteasome-dependent degradation in both U2OS and HeLa cells. This hypothesis is supported by the

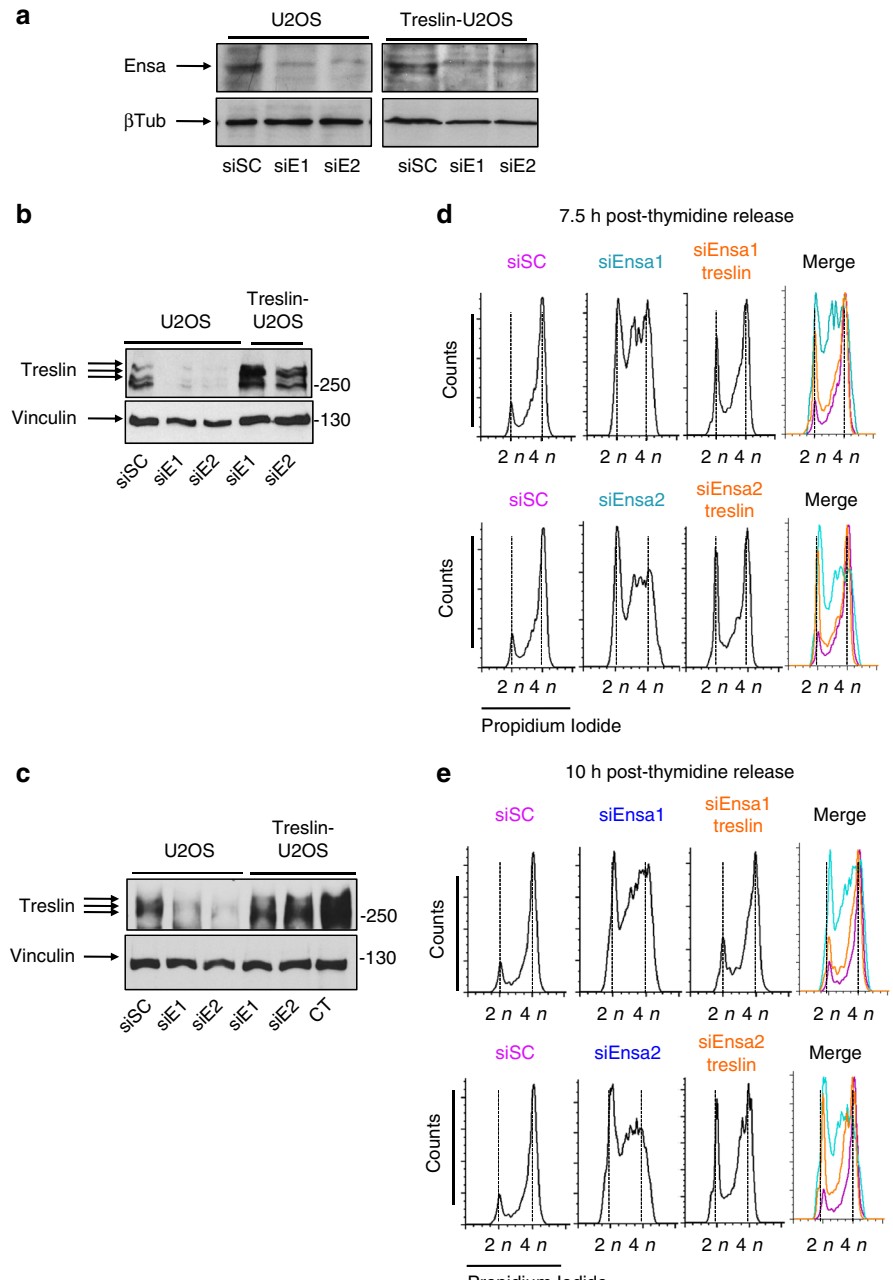

**Fig. 8** Overexpression of Treslin in Ensa siRNA-treated cells rescues normal S phase. **a** Parental or U2OS cells stably overexpressing AcGFP-Flag-Treslin were transfected with scramble (siSC) or Ensa siRNAs #1 (siE1) or #2 (siE2) (for parental) or with Ensa siRNA #1 and #2 (for AcGFP-Flag-Treslin overexpressing cells) and the levels of Ensa and β-tubulin as a loading control, checked by western blot. Cells were synchronised in G1–S by thymidine block 24 h later and recovered for western blot at 7.5 h **b** and 10 h **c** after release. The levels of Treslin in non-transfected U2OS cells stably overexpressing AcGFP-Flag-Treslin were also checked (CT). FACS analysis at 7.5 h **d** and 10 h **e** after release are also shown

fact that the addition to Ensa knocked down cells of either the proteasome inhibitor MG132 or of the NEDDylation activating enzyme inhibitor, MLN4924 prevents the drop of Treslin levels. Interestingly, our findings indicate that the phosphorylation of Treslin by Cdk stabilises this protein, whereas its dephosphorylation by PP2A promotes its proteolysis, suggesting that the Gwl/Ensa pathway controls S-phase length by modulating the PP2A-dependent dephosphorylation and degradation of Treslin.

We do not know which is the physiological function of Treslin instability, however, taking into account the essential role of this protein in replication origin firing, we could hypothesise that its quick degradation in response to massive replicative stress could

rapidly and efficiently delay S-phase progression in response to the intra S-phase checkpoint activation. In addition, the regulation of the levels of this protein could similarly contribute to the establishment of the diverse replication timing programs displayed in different cell types or in differentiated cells[25].

In summary, here we identify a new regulatory mechanism controlling S phase that involves the regulation of Treslin levels by the Gwl/Ensa pathway via the control of the PP2A-dependent dephosphorylation and Cullin-dependent degradation of this protein. This mechanism is essential to modulate replication timing under physiological conditions. Further studies will be required to determine whether this pathway could also participate

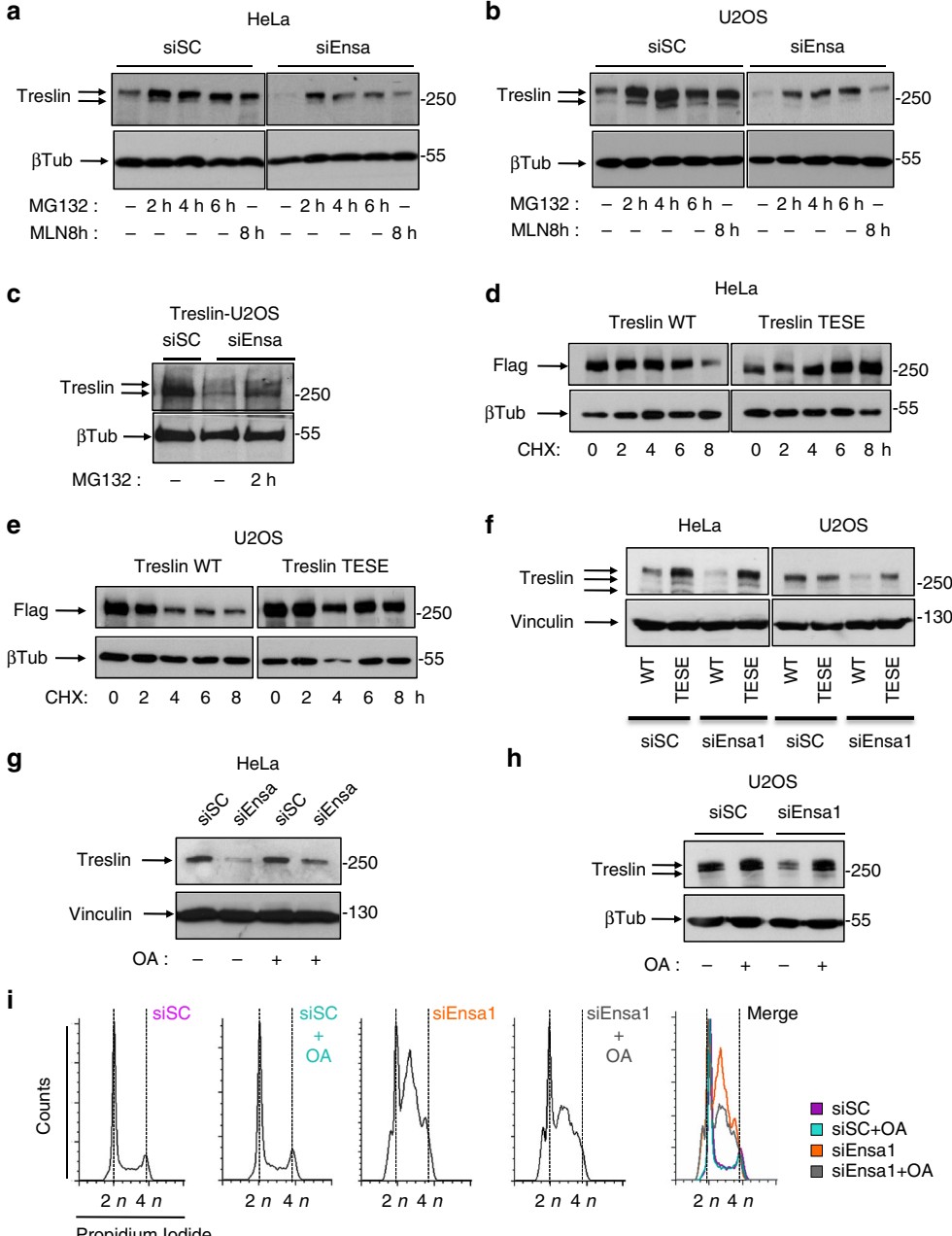

**Fig. 9** Ensa impedes the Cullin-dependent degradation of Treslin by preventing its dephosphorylation by PP2A. **a** Parental HeLa cells were treated with siSC or siEnsa and subsequently incubated for 6 h without (−) MG132, for 2, 4 or 6 h with MG132 (10 μM) or for 8 h with MLN4924 (500 nM) and subsequently recovered for western blot analysis. β-Tubulin levels were used as a loading control. **b** As for **a**, except that U2OS cells were used. **c** U2OS cells stably expressing Treslin cDNA were transfected with siSC or siEnsa and subsequently incubated with (2 h) or without (−) MG132 (10 μM) and Treslin amount checked by western blot. **d** HeLa cells overexpressing a Flag-tagged wild-type (WT) Treslin or the phosphomimetic mutant of this protein T968E/S1000E (TESE) were incubated with cycloheximide for the indicated periods of time and submitted to western blot to measure ectopic Treslin levels. **e** As for **d**, except that U2OS instead of HeLa cells were used. **f** HeLa and U2OS cells were co-transfected with Flag-tagged wild type or the T968/S1000E Treslin mutant and treated with siSC or siEnsa and used for western blot to measure ectopic Treslin levels **g** HeLa cells were transfected with scramble (siSC) and Ensa (siEnsa) siRNAs and 24 h later incubated (+) or not (−) with the PP2A inhibitor okadaic acid (OA; 5 nM). Cells were recovered 24 h later and used for western blot. **h** As for **g**, except that U2OS instead of HeLa cells were used. **i** FACs analysis for HeLa cells in **g** is shown

to the intra-S-phase checkpoint response under pathological conditions.

## Methods

**Cell lines, transfection, infection and synchronisation**. HeLa and U20S cells were obtained from ATCC collection. HeLa PCNA chromobody cells were purchased from Chromotek[22] and U20S cells stably expressing AcGFP-Flag-Treslin were a generous gift of Dr J.F.X. Diffley[31]. All these cells were cultured in DMEM

media supplemented with 10% foetal bovine serum. Stable HeLa cells were generated using PMX-puro Human *Ensa* modified to be resistant to siRNA Ensa1 using site-directed mutagenesis with the oligonucleotide 5′-AACATGGC-CAAAGC**T**AA**A**ATGAAGAA**C**AAA**C**AGCTGCCAAGT-3′ (bold letters correspond to the four mutated nucleotides). Phosphomimetic Treslin T968E S1000E mutant were created from pcDNA5 TO human WT Treslin 3xFlag (gift from W.G. Dunphy) using the Quick change kit (Agilent Technologies) and all mutations were confirmed by DNA sequencing. Oligonucleotides used for site-directed mutagenesis were purchased from Eurogentec (T968E: 5′-

ACTAAGAGTGTGGCCGAGGAACCAGTGCATAAGCAGATC-3′ and S1000E: 5′-ATTGGTGTTGTTGAAGAGGAACCTGAAAAAGGAGATGAAA-3′) (bold letters correspond to the mutated oligonucleotides).

For synchronisation experiments, cells were arrested in G1/S with 2.5 mM thymidine for 24 h. Cells were then washed three times with pre-warmed media, and released from G1/S arrest by adding fresh media. To block cells at the G2/M transition, cells were incubated with 10 μM RO-3306 (Tocris Bioscience). Mitotic cells were obtained by shakeoff after nocodazole treatment (100 ng/ml for 16 h).

Gwl(Lox/Lox) MEFs were grown as described in Alvarez-Fernandez et al.[17] For knockout induction, MEF cells were deprived of serum for 72 h, subsequently transduced with AdenoGFP or AdenoGFP-Cre viruses (gene transfer core vector; University of Iowa, USA) during 5 h and finally stimulated to re-enter the cell cycle with fresh serum-supplemented media. G1/S synchronisation was induced in these cells by aphidicolin (2 μg/ml) for 24 h.

HeLa or U2OS cells were treated with the proteasome inhibitor 10 μM of MG132 for 2, 4 or 6 h (Tocris), or with the NEDDylation activating enzyme inhibitor, MLN4924 (500 nM for 8 h) (Active Biochemicals Co.), with 5 nM of okadaic acid for 24 h, or with cycloheximide (CHX, Sigma) at 40 μg/ml for 2, 4, 6 or 8 h.

siRNAs were synthesised using T7 RiboMaX TM Express RNAi system (Promega) and were transfected with Lipofectamine RNAimax (Life Technologies) according the manufacturer's instructions at a concentration of 50 nM. Double transfection of siRNA and the wild-type or the mutant form of Treslin were performed using Lipofectamine 2000 (Life Technologies) in cells in suspension.

siRNA SC (scramble): 5′-CTTAGCTACGATCAAGTAC-3′
siRNA Ensa #1: 5′-GCCAAGATGAAGAATAAGC-3′ (271–289)
siRNA Ensa #2: 5′-GAAACAAGAAGAAGAGAAC-3′ (9–27).

**Fluorescence-activated cell scanning**. For FACS, cells were fixed with ice-cold 70% ethanol at −20 °C, and washed with phosphate-buffered saline (PBS), DNA was stained with a solution containing propidium iodide (5 μg/ml) and RNAse A (0.5 mg/ml) in PBS. For bivariate FACS analysis, cells were pre-labelled with 10 μM BrdU (Sigma) for 30 min, harvested by trypsinisation and fixed as describe above. Fixed cells were washed once with PBS and incubated with 2 M HCl, 0.5% (vol/vol) Triton X-100 for 30 min, with gentle mixing. After two washes in PBS, cells were incubated 2 h at room temperature with anti-BrdU antibody (Santa Cruz Biotechnology Inc.) and diluted 1/30 in PBS containing 1% BSA, 0.5% Tween 20. Then, BrdU staining was revealed with Alexa Fluor 488-conjugated goat anti-mouse antibody and DNA with a 7AAD (1.5 μg/ml) and RNAse A (0.5 mg/ml) in PBS. Finally, cells were analysed on a FACS Calibur (BD Biosciences) using Cell Quest Pro and Flow Jo software.

**DNA combing**. DNA combing was performed as described in Bialic et al.[32] Briefly, cells were pulsed with IdU (25 μM) for 15 min, then with CldU (200 μM) for 15 min, before proceeding to a thymidine (1 mM) chase for 2 h. After trypsinisation and cell counting, 50,000 cells were resuspended in PBS, pre-warmed at 42 °C, then mixed with an equal volume of 1% low-melting agarose. Agarose plugs were treated overnight at 37 °C with 0.2 mg/ml proteinase K before melting with ß-agarase, DNA combing and signal detection with antibodies against ssDNA (Millipore MAB3034, 1/300), IdU (Becton-Dickinson clone B44, 1:20) and CldU (AbCys clone Bu 1/75) along with appropriate fluorescently labelled secondary antibodies. After image acquisition, fork velocities were calculated on bicolour ongoing forks away from the fibre edge by dividing the length of the second (CldU) tract by the duration of the second pulse (15 min) using 2 kb/μm as stretching rate. Fork density was calculated by dividing the total number of ongoing forks with the total length of DNA fibres, after correction for the percentage of cells in S phase[32].

**Cell extracts and chromatin isolation**. Cells cultured in 35, 60 or 100 mm dishes were rinsed in cold PBS and lysed in 50 mM Tris-HCl, pH 7.5, 120 mM NaCl, 20 mM NaF, 1 mM EDTA, 6 mM EGTA, 15 mM NaPPi, 15 mM pNPP, 1 mM benzamidine, protease inhibitor cocktail, 0.5 mM vanadate, and 1% Nonidet P-40 or in Laemmli buffer (Tris-HCl 10 μM, pH 6.8, SDS 1%, glycerol 5%, β-mercaptoethanol 0.5%, DTT 1 μM) before sonication (30 s). To isolate chromatin, cells were resuspended ($4x10^7$ cells per ml) in buffer A (10 mM HEPES—pH 7.9, 10 mM KCl, 1.5 mM MgCl$_2$, 0.34 M sucrose, 10% glycerol, 1 mM DTT, 1× protease inhibitor cocktail) and processed as described by Méndez and Stillman[33]. Briefly, 0.1% of Triton X-100 was added and cells were incubated 1 min on ice before centrifugation at 1300 × g for 5 min. The supernatant (S1) is then clarified by high-speed centrifugation (10 min, 20,000 × g, 4 °C) to remove cell debris and insoluble aggregates. The new supernatant (S2) contains soluble cytoplasmic proteins. Nuclei were collected in pellet 1 (P1) by low-speed centrifugation (5 min, 1300 × g, 4 °C) after a wash in buffer A without Triton X-100, and were lysed for 30 min on ice in buffer B (3 mM EDTA, 0.2 mM EGTA, 1 mM DTT and 1× protease inhibitor cocktail). Insoluble and chromatin enriched fraction (P3) was collected by centrifugation (5 min, 1700 × g, 4 °C), washed once in buffer B and centrifuged again under the same conditions. The final chromatin pellet was resuspended in Laemmli sample buffer and sonicated for 20 s. The first supernatant (S3) contains soluble nuclear fraction.

**Antibodies and immunoblots**. Proteins (20–50 μg) were loaded on a polyacrylamide gel and then transferred onto Immobilon-P. Membranes were incubated with homemade rabbit polyclonal antibodies raised against recombinant full length 6His-human Ensa protein (anti-hEnsa) (1/1000), phospho-Arpp19/Ensa S67/S71 (Cell Signaling, #5240, 1/250,), human Greatwall (anti-hGreatwall) (1/1000), phospho-Greatwall S875[34], α-Tubulin (1/1000 clone C102, generous gift of J.M. Andreu, Spain), Treslin (1/1000, Bethyl Laboratories Inc., A303-472A), phospho-Ser CDK substrate (1/1000, Cell Signalling, #9477S), Histone H3 (1/500, Santa Cruz, Biotechnology Inc., SC-FL136), Cdk1 (1/1000, Santa Cruz, Biotechnology Inc., SC-954), MCM-2 (1/10000, Abcam, ab4461), MCM-3 (1/2500, Abcam, ab4460), MCM-4 (1/5000, Abcam, ab4459) and MCM-5 (1/5000, Abcam, ab17967). Mouse monoclonal used are MCM-7 antibody (1/1000, Abcam, ab2360), Flag M2 (1/500, Sigma, F3165), Vinculin (1/1000, Sigma,V9131 Hvin-1) and β-Tubulin (1/500, clone E7, generous gift from N. Morin). Uncropped scans of all the western blot membranes are supplied in the Supplementary Information file.

**Immunoprecipitation**. Cells were lysed in NETN buffer (0.25% IGEPAL, 150 mM NaCl, 50 mM Tris and 1 mM EDTA) complemented with 500 nM microcystin-LR. Immunoprecipitations were carried out for 1 h at room temperature using 400 μg of protein and 2 μg of Ensa polyclonal antibody affinity purified against 6His-human Ensa protein immobilised on protein A (Dynal Beads, Invitrogen). Immunoprecipitates were then analysed by immunoblotting as described above.

**EdU/BrdU labelling**. Cells were grown on glass coverslips and pulsed with 10 μM EdU and/or 10 μM BrdU for 30 min. For EdU/BrdU kinetics, cells were grown in media containing 5 μM thymidine between the two stainings. Cells were then fixed for 15 min in PBS with 3.2% formaldehyde and permeabilised in 0.2% Triton X-100 for 10 min, then treated with 4 N HCl for 20 min. All cells were washed and blocked (3% BSA, 0.1% Tween 20 in PBS) for 45 min. Click-iT- EdU Alexa 530 staining was performed as per manufacturer instruction (Invitrogen). After three washes in PBS, cells were incubated with monoclonal anti-BrdU antibody (1/400, MoBu-1 Exbio, #11-286-100), and revealed by an Alexa Fluor 633 conjugated goat anti-mouse antibody. Coverslips were mounted using Prolong Gold mounting medium (Invitrogen) and captured using a confocal Leica SP5-SMD microscope, with a Leica ×63/APO 1.4 lens, powered by LASAF software. Serial 0.5 μm Z-sections were taken, and maximum projections were generated by using Image J software.

**Immunohistochemistry**. Cells were fixed as above, permeabilised in 0.2% Triton X-100 and incubated with rabbit polyclonal anti-hEnsa (1/100), anti-hGreatwall (1/100) and monoclonal anti-BrdU. Antibodies were revealed with anti-Rabbit Alexa Fluor 546 (Life Technologies, A11035, 1/1000), anti-Mouse Alexa Fluor 546 (Life Technologies, A11018, 1/1000), anti-Rabbit Alexa Fluor 488 (Life Technologies, A11070, 1/1000) and anti-Mouse Alexa Fluor 488 (Life Technologies, A11017, 1/1000).

Alexa Fluor 546-conjugated or Alexa Fluor 488-conjugated goat anti-rabbit or mouse antibodies (Molecular Probes, Life Technologies). Cells were analysed as described previously[35]. Images were taken with a Leica SP5 confocal microscope or with a MetaMorph-driven software (Molecular Devices, Sunnyvale, CA) wide-field Axioimager Z2 fluorescence microscope (Zeiss, Jena, Germany) with a PL APO ×63 or ×40 objectives (numerical aperture 1.32; Leica, Melville, NY) and a CoolSNAP HQ2 3 camera (Photometrics, Woburn, MA). For quantification experiments, cells were acquired using scan slide module and analysed with ImageJ. Images were then processed using Photoshop (Adobe, San Jose, CA).

**Live imaging**. HeLa PCNA chromobody cells (Chromotek)[22] were transfected with SC siRNA or Ensa siRNA during 24 h, then blocked in G1/S with thymidine for 24 h, and released in fresh medium before analysis by time-lapse microscopy. Time-lapse confocal microscopy was performed using a spinning disk CSU-X1 Andor coupled to an NIKON inverted microscope. Images were acquired with 20 Z-section at 0.7 μm steps, taken every 5 min for up to 72 h with a ×40/1.6 lens. Maximum projections were generated using Image J.

**RT-PCR and real-time quantitative PCR**. RNAs from cell lines were isolated using the RNeasy Mini kit (Qiagen). Five micrograms of total RNA was reverse transcribed using oligodT (Roche) in a final volume of 20 μl using Superscript III (Invitrogen). After reverse transcription, 100 ng of cDNA was used for real-time quantitative PCR, performed with a Lightcycler and the SYBR Green fast start kit (Roche, Germany). Primers sequences used in real-time PCR specific for human TICRR (Treslin) genes as well as for the housekeeping gene MRPL19 were as follows:

HumTreslin-F: 5′-AAACACTTTGGATTCGGAGGT-3′;
HumTreslin-R 5′- TTCTGCAGCCTTTCTGGAGT-3′;
HumMRPL19-F: 5′-GGGATTTGCATTCAGAGATCAG-3′;
HumMRPL19-R: 5′-GGAAGGGCATCTCGTAAG-3′.

**Data availability**. The data that support the findings of this study are available form the corresponding author upon reasonable request.

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

## Acknowledgements

We are grateful to Dr J.F.X. Diffley (Francis Crick Institute, London, UK), Dr W.G. Dunphy (Caltech, Pasadena, USA) and Dr M. Malumbres (CNIO, Madrid, Spain) for generously supplying U2OS cells stably expressing AcGFP-Flag-Treslin, pcDNA5-Treslin vector and Mastl(Lox/Lox) MEFs, respectively. We are also indebted to Dr J.M. Andreu (CIB, Madrid, Spain) and Dr N. Morin (CRBM-CNRS, Montpellier, France) for anti-αTubulin (clone C102) and anti-β-tubulin (clone E7) monoclonal antibodies, respectively. We thank S. Lachambre, V. Georget and S. De Rossi from Montpellier RIO Imaging for microscopy facilities, M. Boyer-Clavel for FACS facility and M. Drac (Montpellier DNA Combing facility) for DNA combing and analysis. We are grateful to Dr S. Ovejero and Dr A. Constantinou (IGH, Montpellier, France) for the generous gift of anti-MCM2-7 antibodies. We are indebted to V. Coulon (IGMM, Montpellier, France) for helpful hints and discussions and to Y. Perez and A. Barral for their precious help on some key experiments. This work was supported by the Ligue Nationale Contre le Cancer (Equipe Labellisée), the 'Agence Nationale de la Recherche' (ANR-10-BLAN-1207), the 'Fondation pour la Recherche Medicale' (FRM) 'Equipes FRM 2015' and the 'Fondation ARC pour la Recherche sur le Cancer' (PJA 20141201679). K.H. is a Labex EpiGenMed program fellowship (ANR-10-LABX-12-01).

## Author contributions

A.B. made the first initial observations on S-phase phenotype by Ensa knockdown that are depicted in Fig. 1a and set up time-lapse experiments with PCNA-targeting chromobody cell lines. A.G.-A. obtained data showed in Fig. 3b and part of data of Fig. 4. J.V. performed experiment shown in Fig. 6c. K.H. performed experiments for Fig. 6a and P.R. participated to the set up of all real-time quantitative PCR. E.S. performed all DNA combing experiments of Fig. 3a. S.C. performed all other experiments. T.L. and A.C. designed the experiments with the person(s) performing them and wrote the paper. All authors approved the manuscript.

## Additional information

**Competing interests:** The authors declare no competing financial interests.

