## [Peer Review File · Nature Communications]

Reviewers' Comments:

Reviewer #1 (Remarks to the Author)

This is an interesting but somewhat puzzling paper. It is now well established that the Gwl (Greatwall kinase) - Ensa/Arpp19 - PP2A-B55 pathway plays an essential role for full activation of cyclin B-Cdk1 and its substrates phosphorylation in M-phase. In the present study, the authors examined a role for Ensa in regulating S-phase progression. They found that Ensa in which the Gwl site is phosphorylated is detectable in the nuclei in S-phase; Ensa depletion by siRNA causes elongation in whole length of S-phase ; and this effect is caused by the SCF-dependent decrease in Treslin protein levels through a general decrease of the firing of early, mid and late replication origins. The authors concluded that Ensa controls the ubiquitin-proteasome dependent degradation of Treslin and hence, regulates S-phase duration. So far I know, this is the first report that shows involvement of Ensa in regulating DNA replication.

Although these findings look very stimulating and of interest to the wide field of cell biology, many questions, particularly mechanistic issues, remain unanswered as indicated below. I think the impact of the present study would be strengthened much more, if any data could be added to provide mechanistic insights.

Major comments

1. Roles for Gwl ---- I am wondering whether Gwl is indeed involved in the present defect caused by Ensa depletion. Could explain the reason why the extent of S-phase accumulation was much less in Gwl-knocked out cells than in Ensa-knocked down cells (Fig. 5). Further, could possibly show whether the present defect caused by the Gwl knockout is restored by expression of mutant Ensa, in which the Gwl phosphosite is altered to phosphomimetic residue, but not by expression of wild type Ensa. I think the present observations simply indicate the parallel occurrence of phosphorylated Gwl and Ensa phosphorylated on the Gwl site (Fig. 6), but not causal involvement of the Gwl-Ensa pathway (end of the Abstract; p.15 - line 360; Fig. 8d).
2. Roles for B55 ---- I am wondering why siB55 increased, but not restored, the defect (i.e., decrease in Treslin protein levels) caused by siEnsa (Fig. 7f). Although this "increase" has not been mentioned at all, could mention whether this is significant, and if so, how this could be understood. In any case, I am wondering whether phosphorylation of Treslin by Cdk1 (Sansam et al., 2015) has any implication in the present effect of Ensa.
3. Treslin degradation ---- I am wondering whether the SCF-dependent degradation of Treslin in Ensa-depleted cells occurred in interphase (or S-phase) but not in M-phase. Could show this.
4. Arpp19 ---- Although Ensa was studied in the present study, Arpp19 has not been addressed at all. Could mention whether the cells used here (HeLa, U2OS, and MEF) contain Arpp19. If so, could mention how the authors evaluate the present observations obtained in the presence of Arpp19.

Minor comments

1. Effect of R-Ensa ---- Could explain why "siSC" and "siEnsa + R-Ensa" do not show the similar pattern in Fig. 2d.
2. Confusing naming ---- siEnsa1 and siEnsa2 (and siEnsa3) appear both in Fig. 1 and Fig. 4c. However, what is indicated by these numberings is different between these figures (i.e., different sequences in Fig. 1, but experimental numbers in Fig. 4c).

3. Phosphosite on Ensa by Gwl ---- Could make clear whether it is S62/S67 (p.10 - line 228) or S71 (p.29 - line 715).

4. Fig. 6d ---- Fig. 6d does not look to fit the title of Fig. 6. Why not is Fig. 6d moved to the first figure in Fig. 7?

5. MG132 treatment ---- What is meant by (-) in Fig. 7c-e? Is this just before addition of MG132 (i.e., 0 hr), or 2, 4, or 8 hr culture in the absence of MG132?

Reviewer #2 (Remarks to the Author)

In this interesting study the authors investigate the role of ENSA in the regulation of cell cycle progression using HeLa and U2OS cells. ENSA has previously been described as a substrate of the Greatwall kinase, which in its phosphorylated state inhibits mitotic PP2A activity, thereby stabilising the mitotic state. The current paper identifies a novel and unexpected role for ENSA and its regulator the Greatwall kinase in determining S phase length by regulating treslin protein levels.

The paper shows that knockdown of ENSA or Greatwall results in delayed S phase progression due to a decrease in origin firing. Both ENSA and Greatwall were found to be active and localised to the nucleus during S phase. The downstream targets of the Greatwall-ENSA pathway during S phase were investigated. Unexpectedly, ENSA knockdown resulted in SCF-mediated degradation of treslin, a key CDK substrate required for the initiation of DNA replication. This treslin degradation was independent of PP2A subunit B55. Significantly, treslin overexpression rescued the S phase phenotype of ENSA knockdown suggesting that the primary S phase target of ENSA is treslin.

Overall we support the publication of this paper: the experimental design is logical and appropriate to answer the questions the authors aim to address. However, whilst the FACs data presented are on the whole convincing, in general the immunoblot data is of an inconsistent quality and in many cases the data would benefit from quantification and statistical analyses. In order to further support the authors' hypothesis the following should be addressed prior to acceptance for publication:

1. Figure 3b addresses whether ENSA knockdown impairs origin licensing by the rather indirect method of performing knockdown during G1 when licensing should already have occurred. In order to eliminate any concern that ENSA knockdown limits replication licensing the level of licensing in both control and ENSA knockdown should be determined by immunoblotting of chromatin-bound Mcm proteins.

2. Figures 7 and 8 address the regulation of treslin protein levels by ENSA, but the quality of immunoblotting in these figures is very inconsistent (especially of U2OS cell extracts). In order to support the conclusions made treslin protein levels in control, ENSA knockdown (using both siRNAs) and Greatwall kd cells should be compared by immunoblotting to that of treslin knockdown samples in both HeLa and U2OS cells. This data would replace current figure 7a; in the current version of this data, both siENSA samples appear to be underloaded. Quantification of this data would allow direct comparison with the RNA data in figure 7b. See below for more specific comments on individual parts of figures 7 and 8.

Minor comments:

3. Fig 1a: the arrow points to what appears to be one band but both bands are reduced upon siENSA: please clarify which band(s) is (are) Ensa?

4. Fig 1b: this blot should be repeated ,as presented no conclusion can be made due to poor recognition of ENSA in the control sample.

5. Fig 2c: the phenotypic data should be matched by an immunoblot to quantify ENSA levels in each treated sample.
6. Sup fig 1a: the immunoblot is unconvincing as a positive signal in the control cells is not clear.
7. Fig 2d and Sup 1b: Please clarify the difference between siENSA R-ENSA in fig 2d and sup 1b; S phase progression is clearly faster in fig 2d.
8. Sup fig 2c: The graph segments E/E, E/M, E/L etc do not match the legend (E, M and L) - the graph segments should be described explicitly in the legend.
9. The text in line 241 is not correct as ENSA levels on chromatin have not been determined.
10. Fig 7c: As presented, this reviewer remains unconvinced that treslin accumulates with MG132 treatment in the ENSA kd cells: these blots should be repeated and quantified.
11. Fig 7d, e: The treslin blots are not very convincing and are different to others derived from U2OS cell extracts. They should be repeated.
12. Fig 8a: The treslin-U2OS siE1 siE2 appear underloaded relative to the other lanes.
13. Fig 8b, c: Loading controls in these blots is unconvincing - equal loading of these lanes is paramount.
14. On line 47 a reference to Sansam et al (Genes Dev 2010, 24, 183-94) should be added.
15. Line 80 missing reference to Fig 1b.

Reviewer #3 (Remarks to the Author)

In this manuscript, the authors use siRNA to knockdown levels of Ensa. The result is a substantially prolonged S phase. The authors go on to show that Treslin levels are decreased as a result of siRNA knockdown of Ensa. The decrease in Treslin levels is not the result of transcriptional changes, but it relates to the enhanced proteasomal degradation of Treslin as a result of siRNA knockdown of Ensa. Furthermore, the authors go on to show that overexpression of Treslin restores S phase to its normal length.

Overall, this is a nice story with thorough and rigorous analysis. The data clearly point to a new pathway for Treslin degradation. My only problem with this manuscript is that I am not sure that it is suitable for the broad readership of Nature Communications, since the observed findings are very specialized. It is already known that decreasing Treslin levels prolong S phase (Sanson et. al.). The only new information is that Ensa regulates proteasome degradation of Treslin. As a minor point, the discussion of Treslin function is incomplete. There are recent papers demonstrating that Treslin binds not only to TopBP1, but also to Mcm2-7, single-stranded DNA, and the Dbf4-Cdc7 kinase (Kaplan et al). These papers should be discussed as well.

POINT BY POINT ANSWER

Referee 1:

- 1. Roles for Gwl ---- I am wondering whether Gwl is indeed involved in the present defect caused by Ensa depletion. Could explain the reason why the extent of S-phase accumulation was much less in Gwl-knocked out cells than in Ensa-knocked down cells (Fig. 5).**

We calculated the ratio of S phase cells in Gwl knockout (KO) MEFs versus CT MEFs and of siSC versus siENSA in HeLa and U2OS cells. We observed a ratio of 1,57 in MEFs and 2 in HeLa and U2OS cells. These differences can be explained, at least in part, by the mitotic phenotypes induced by Gwl KO. Gwl KO results in important cytokinesis defects and subsequent formation of polyploidy cells. These cells are delayed or blocked in mitosis and would partially decrease S phase cell accumulation in Gwl KO MEFs. In addition, polyploidy cells performing S phase will not be detected by FACs analysis, since they will display more than 4n DNA content and they will be moved to the right in the X axis of the FACs analysis. Together, these two issues likely participates to the underestimation of the number of cells performing S phase after Gwl KO.

- 2. Further, could possibly show whether the present defect caused by the Gwl knockout is restored by expression of mutant Ensa, in which the Gwl phosphosite is altered to phosphomimetic residue, but not by expression of wild type Ensa.**

Unfortunately, the phosphomimetic form of Ensa in the Gwl site does not rescue mitosis in Gwl-depleted *Xenopus* egg extracts (our unpublished results), indicating that it does not act as the phosphorylated protein. As so, we cannot use it to restore Gwl knockout defect in MEF cells. However, our new results demonstrate that Treslin phosphorylation by Cdk stabilises this protein (Fig. 9c) and that the inhibition of the phosphatase PP2A results in a significantly restoration of S phase length in siEnsa treated cells (Fig. 9d). In addition, we show that Gwl KO also promotes the degradation of Treslin (Fig.7d). Together all these data supports the hypothesis that Gwl through the phosphorylation of Ensa promotes the inhibition of PP2A and stabilises Treslin during S phase.

3. Roles for B55 ---- I am wondering why siB55 increased, but not restored, the defect (i.e., decrease in Treslin protein levels) caused by siEnsa (Fig. 7f). Although this "increase" has not been mentioned at all, could mention whether this is significant, and if so, how this could be understood.

In any case, I am wondering whether phoshorylation of Treslin by Cdk1 (Sansam et al., 2015) has any implication in the present effect of Ensa.

We thank the referee for this comment. When we deeper analysed the effect of Cdk-dependent phosphorylation of Treslin half-life (Fig. 9c, revised version), we realised that Cdk phosphorylations conferred stability to this

protein. We subsequently measured the effect of the inhibition of the PP2A phosphatase with low doses of okadaic acid in siEnsa treated cells (Fig. 9d). We observed a significant rescue of the S phase extension induced by Ensa knockdown. In addition, the incubation of siEnsa treated cells with okadaic acid mostly restored Treslin levels. From all these results we concluded that Gwl/Ensa pathway is controlling Treslin levels and S phase progression by inhibiting PP2A-dependent dephosphorylation of this protein.

As underlined by this referee, we observed most of the times a much important decrease of the Treslin levels in Ensa-B55 double siRNA treated cells. In addition, when double siRNA was performed, we repetitively observed a decrease of the tubulin levels in our western blot analysis although similar amounts of protein from cell lysates were loaded in the SDS PAGE gels.

Taking into account our new findings on the role of Cdk-dependent phosphorylation of Treslin in Treslin stability, we interpret that the effects induced by double siRNA were probably induced by toxicity.

New data on the role of Cdk1-dependent phosphorylation on Treslin stability is included in this new version of the manuscript and double Ensa-B55 siRNA data has been excluded.

4. Treslin degradation ---- I am wondering whether the SCF-dependent degradation of Treslin in Ensa-depleted cells occurred in interphase (or S-phase) but not in M-phase. Could show this.

We checked Treslin degradation in S phase and M phase synchronised

siEnsa treated cells. Our data indicate that in both phases of the cell cycle the depletion of Ensa induces Treslin degradation. This data is included in Fig 7e.

- 5. Arpp19 ---- Although Ensa was studied in the present study, Arpp19 has not been addressed at all. Could mention whether the cells used here (HeLa, U2OS, and MEF) contain Arpp19. If so, could mention how the authors evaluate the present observations obtained in the presence of Arpp19.**

HeLa and U2OS cells display very low levels of Arpp19. In addition, although both proteins, when ectopically added to interphase *Xenopus* egg extracts, promote mitotic entry, it is not clear yet whether the endogenous proteins perform identical roles. However, even if Arpp19 participates to inhibit PP2A towards Treslin dephosphorylation, the levels of Arpp19 are so low in HeLa and U2OS cells compared to Ensa that it is likely that they would not be sufficient to assure the complete inhibition of PP2A and thus the stabilisation of Treslin.

Minor Points

- 1. Effect of R-Ensa ---- Could explain why "siSC" and "siEnsa + R-Ensa" do not show the similar pattern in Fig. 2d.**

Our different experiments on R-Ensa cells demonstrate that they display a similar cell cycle pattern than control cells as displayed in Supplementary Fig. 1 (new version). We thus, changed Fig. 2d for a best representative figure.

- 2. Confusing naming ---- siEnsa1 and siEnsa2 (and siEnsa3) appear both in Fig. 1 and Fig. 4c. However, what is indicated by these numberings is different between these figures (i.e., different sequences in Fig. 1, but experimental numbers in Fig. 4c).**

siEnsa1, siEnsa2 and siEnsa3 naming in Fig. 4c has been changed by Cell1, Cell2 and Cell3.

- 3. Phosphosite on Ensa by Gwl ---- Could make clear whether it is S62/S67 (p.10 - line 228) or S71 (p.29 - line 715).**

Gwl phosphosites in Ensa and Arpp19 have been clarified in the text.

- 4. Fig. 6d ---- Fig. 6d does not look to fit the title of Fig. 6. Why not is Fig. 6d moved to the first figure in Fig. 7?**

Fig 6d has been moved to Fig. 7a.

- 5. MG132 treatment ---- What is meant by (-) in Fig. 7c-e? Is this just before addition of MG132 (i.e., 0 hr), or 2, 4, or 8 hr culture in the absence of MG132?**

MG132 (-) corresponds to cells cultured in the absence of this proteasome inhibitor. This point has been clarified in Figure legends.

Referee 2

- 1. Figure 3b addresses whether ENSA knockdown impairs origin licensing by the rather indirect method of performing knockdown during G1 when licensing should already have occurred. In order to eliminate any concern that ENSA knockdown limits replication licensing the level of licensing in both control and ENSA**

knockdown should be determined by immunoblotting of chromatin-bound Mcm proteins.

Chromatin bound MCMs in siEnsa treated cells has been determined by immunoblotting and compared to siSC treated cells. Results are shown in Fig 3c.

- 2. Figures 7 and 8 address the regulation of treslin protein levels by ENSA, but the quality of immunoblotting in these figures is very inconsistent (especially of U2OS cell extracts). In order to support the conclusions made treslin protein levels in control, ENSA knockdown (using both siRNAs) and Greatwall kd cells should be compared by immunoblotting to that of treslin knockdown samples in both HeLa and U2OS cells. This data would replace current figure 7a; in the current version of this data, both siENSA samples appear to be underloaded. Quantification of this data would allow direct comparison with the RNA data in figure 7b. See below for more specific comments on individual parts of figures 7 and 8.**

Treslin Immunoblots in HeLa, U2OS, and U2OS overexpressing Treslin have been repeated and are shown in Fig. 7b. We also showed in this new version of the manuscript that Gwl knockout MEFs display a decrease of Treslin levels strengthening the hypothesis of a role of the Gwl/Ensa pathway in the control of Treslin levels during S phase.

Minor Comments:

- 3. Fig 1a: the arrow points to what appears to be one band but both bands are reduced upon siENSA: please clarify which band(s) is (are) Ensa?**

Bands corresponding to Ensa protein have been clarified in Fig. 1a.

- 4. Fig 1b: this blot should be repeated ,as presented no conclusion can be made due to poor recognition of ENSA in the control sample.**

Western blot in Fig. 1b has been repeated.

- 5. Fig 2c: the phenotypic data should be matched by an immunoblot to quantify ENSA levels in each treated sample.**

Unfortunately, we don't have any sample available of these cells, since they were directly submitted to time-lapse microscopy. We could repeat this curve-dose response to measure Ensa levels, but it would not correspond to the same experiment. If necessary, we could exclude this Figure from the manuscript, since the effect of siEnsa on S phase has been clearly shown by different other types of experiments.

- 6. Sup fig 1a: the immunoblot is unconvincing as a positive signal in the control cells is not clear.**

Western blot in Supplementary Fig. 1a has been replaced.

- 7. Fig 2d and Sup 1b: Please clarify the difference between siENSA R-ENSA in fig 2d and sup 1b; S phase progression is clearly faster in fig 2d.**

Fig. 2d has been changed for a most representative FACs analysis of the results obtained for this experiment.

8. **Sup fig 2c: The graph segments E/E, E/M, E/L etc do not match the legend (E, M and L) - the graph segments should be described explicitly in the legend.**

Figure Legend for supplementary Fig.2 has been modified.

9. **The text in line 241 is not correct as ENSA levels on chromatin have not been determined.**

Text on Ensa levels in the nuclei has been corrected.

10. **Fig 7c: As presented, this reviewer remains unconvinced that treslin accumulates with MG132 treatment in the ENSA kd cells: these blots should be repeated and quantified.**

Experiments checking the effect of MG132 and MLN4924 in Treslin stability have been repeated and the results are shown in Fig. 9a. Quantification can be provided if the referee considers that is required.

11. **Fig 7d, e: The treslin blots are not very convincing and are different to others derived from U2OS cell extracts. They should repeated.**

The new western blots are shown in Fig.9a

12. **Fig 8a: The treslin-U2OS siE1 siE2 appear underloaded relative to the other lanes.**

Treslin/ β Tubulin ratios in U2OS siEnsa1 and siEnsa2 quantification in Fig.8a

13.Fig 8b, c: Loading controls in these blots is unconvincing - equal loading of these lanes is paramount

Treslin/Vinculin ratios in U2OS cells overexpressing or not Treslin Fig.8 b and c

As the referee can see in the quantifications of all these western blots, although samples were underloaded in some cases, the ratios are in accord

with the results described in the text. We could introduce these quantifications in the manuscript if this referee considers that they are required.

14. On line 47 a reference to Sansam et al (Genes Dev 2010, 24, 183-94) should be added.

Sansam et al. reference has been added.

15. Line 80 missing reference to Fig 1b.

Reference to Fig.1b has been added.

Referee 3

1. Overall, this is a nice story with thorough and rigorous analysis. The data clearly point to a new pathway for Treslin degradation. My only problem with this manuscript is that I am not sure that it is suitable for the broad readership of Nature Communications, since the observed findings are very specialized. It is already known that decreasing Treslin levels prolong S phase (Sanson et. a.l). The only new information is that Ensa regulates proteasome degradation of Treslin.

We don't agree this referee's opinion. Although it is known that Treslin levels are essential to control S phase progression, nothing was known about the mechanisms regulating the amount of this protein. We identified here the first mechanism controlling Treslin levels as a Cullin and proteasome-dependent ubiquitination and degradation. We also show that

this ubiquitination is regulated by Cdk-dependent phosphorylation and PP2A-dependent dephosphorylation. We finally demonstrate that this pathway is controlled by the Gwl/Ensa pathway. These results open new important questions and demonstrate a new unsuspected role of the Gwl/Ensa/PP2A pathway out of mitosis in the control of the S phase.

2. As a minor point, the discussion of Treslin function is incomplete.

There are recent papers demonstrating that Treslin binds not only to TopBP1, but also to Mcm2-7, single-stranded DNA, and the Dbf4-Cdc7 kinase (Kaplan et al). These papers should be discussed as well.

Papers demonstrated Treslin binding to MCMs and Dbf4-Cdc7 have been commented and the corresponding references included.

Reviewers' Comments:

Reviewer #1 (Remarks to the Author)

I am somewhat surprised to find that a major suggestion in the present study has been reversed: in the revision the authors suggest that control of Treslin stability by Gwl/Ensa is mediated by PP2A inhibition, whereas in the previous version they suggested that it is not mediated by PP2A. I understand this change and also most of rebuttal comments. However, as indicated below, I still feel some difficulty to follow the data in Fig. 9, and would like to suggest the authors to reinforce their revised suggestion.

1. I suppose that all cells used in Fig. 9 were not synchronized but randomly cultured. FACs analysis is presented only in Fig. 9d, and hence I am wondering what happened in the cell cycle progression after addition of MG132 (Fig. 9a) or CHX (Fig. 9c). I think that the cell cycle arrest occurred at various stages after addition of these drugs, and particularly, the arrest stage may be different between siSC and siEnsa. Could explain how the authors evaluate the difference in Treslin levels between experimental groups and control groups.

I think it may help the readers if the authors could present, as basic information, the dynamics of Treslin protein levels and of Treslin phosphorylation levels on Cdk sites through normal cell cycle progression, although this is not an essential request.

2. In Fig. 9a, I am still wondering what is meant by "MG132 (-)" in the most left lanes both of siSC and siEnsa. 0 hr incubation? or how many hours incubation in the absence of MG132?

3. In Fig. 9c, I am wondering why protein synthesis should have been inhibited by CHX. Could explain the reason for this.

In addition, could possibly further present the data on the fate of both Flag-Treslin WT and Flag-Treslin TESE after siEnsa.

4. Looking at Fig. 9c and d, I am wondering whether phosphorylation levels of Treslin on the T968 and S1000 Cdk sites are actually decreased in siEnsa cells. If the authors could possibly show it, I think the present paper would become more convincing.

5. In Fig. 9d, the FACs profile appears to represent U2OS (but not HeLa). Could comment on it in the legend.

Reviewer #2 (Remarks to the Author)

The revised version of the manuscript greatly improves the technical problems in the original. I now support publication.

Reviewer #3 (Remarks to the Author)

I believe the authors have done an excellent job of addressing all of the concerns of the reviewers, and the manuscript is now suitable for publication in Nature Communications.

POINT BY POINT ANSWER

Referee 1:

I am somewhat surprised to find that a major suggestion in the present study has been reversed: in the revision the authors suggest that control of Treslin stability by Gwl/Ensa is mediated by PP2A inhibition, whereas in the previous version they suggested that it is not mediated by PP2A. I understand this change and also most of rebuttal comments. However, as indicated below, I still feel some difficulty to follow the data in Fig. 9, and would like to suggest the authors to reinforce their revised suggestion.

1. I suppose that all cells used in Fig. 9 were not synchronized but randomly cultured. FACs analysis is presented only in Fig. 9d, and hence I am wondering what happened in the cell cycle progression after addition of MG132 (Fig. 9a) or CHX (Fig. 9c). I think that the cell cycle arrest occurred at various stages after addition of these drugs, and particularly, the arrest stage may be different between siSC and siEnsa. Could explain how the authors evaluate the difference in Treslin levels between experimental groups and control groups. I think it may help the readers if the authors could present, as basic information, the dynamics of Treslin protein levels and of Treslin phosphorylation levels on Cdk sites through normal cell cycle progression, although this is not an essential request.

The effect of MG132 and CHX in cell cycle progression in siSC and siEnsa treated cells has been analysed and shown in Supplementary Fig. 4 and Supplementary Fig. 5. The addition of 10 μ M MG132 did not induce a noticeable effect either in parental, siSC or siEnsa treated HeLa cells. In U2OS a small delay in S phase was observed in a similar extent in all treated and not treated cells, although in siEnsa cells, this delay accumulated to the one induced by Ensa knockdown (Supplementary Fig. 4). No effect of CHX in cell cycle progression was observed (Supplementary Fig. 5). This information is added in the main text: lanes 305-310 and 329-330.

Dynamic of Treslin protein levels throughout the cell cycle has been added in Supplementary Fig. 3. Our data show that endogenous Treslin levels are high in M/G1, decrease during S phase, increase during G2 and reach maximum levels at mitosis. These results have been added in the main text: lanes 267-272.

Concerning phosphorylation levels on Cdk sites, unfortunately, we do not have any phospho-antibody available against these sites and at least from three to four months would be required to obtain these antibodies to answer this concern. In addition, only one phospho-antibody against S1000 Cdk site of Treslin has been described in the bibliography (Kumagai et al. JCB 2011). However, although authors focused their interest in the phosphorylation of endogenous Treslin in U2OS cells, this antibody was exclusively used for ectopic overexpressed protein in vitro phosphorylated

by purified Cdk kinase, suggesting that it was not able to recognise endogenous phosphorylated Treslin.

- 2. In Fig. 9a, I am still wondering what is meant by “MG132 (-)” in the most left lanes both of siSC and siEnsa. 0 hr incubation? or how many hours incubation in the absence of MG132?**

In Fig. 9a, MG132(-) cells were not treated with MG132 and recovered at the end of the experiment after 6 h-incubation. This information has been pointed out in the main text, Figure Legend 9, lane 675.

- 3. In Fig. 9c, I am wondering why protein synthesis should have been inhibited by CHX. Could explain the reason for this. In addition, could possibly further present the data on the fate of both Flag-Treslin WT and Flag-Treslin TESE after siEnsa.**

In Fig.9c, cells are transiently transfected with vectors with constitutive promoters that induce a quite high and continuous synthesis of ectopic Treslin. Under these conditions and in the absence of Ensa knockdown, endogenous proteolysis rate is not sufficient to uncover ectopic Treslin synthesis and the addition of a protein synthesis inhibitor is required to visualize Treslin proteolysis. This point has been clarified in the text (lanes 328-329).

The stability of WT and Flag-Treslin TESE after siEnsa has been analysed and the new data added in Fig. 9d and in the main text lanes 334-335. As for

parental cells, the TESE Treslin mutant is stable under siEnsa treatment whereas the wild type form is degraded.

4. **Looking at Fig. 9c and d, I am wondering whether phosphorylation levels of Treslin on the T968 and S1000 Cdk sites are actually decreased in siEnsa cells. If the authors could possibly show it, I think the present paper would become more convincing.**

Unfortunately, as pointed out before, we have no phospho-Treslin antibodies available and the only phospho-antibody described in the bibliography does not likely recognise endogenous phosphorylated Treslin.

5. **In Fig. 9d, the FACs profile appears to represent U2OS (but not HeLa). Could comment on it in the legend.**

FACs profile shown in Fig. 9e (Fig. 9d in the previous manuscript) corresponds to HeLa cells. This information has been commented in the Figure legend.

Reviewers' Comments:

Reviewer #1:

Remarks to the Author:

I think the authors have re-revised the manuscript almost adequately. I now support publication of this paper. Before publication, however, I would like to suggest the authors to improve the figure presentation as follows.

Looking at the newly added Supplementary Figure 3 using U2OS cells, I have realized that Treslin bands after immunoblots are not so clear in other figures mostly using HeLa cells. Could indicate which band(s) is (are) Treslin (upper, lower, or both band(s) ?) particularly in Figure 7e and 7b/HeLa.

POINT BY POINT ANSWER

Referee 1:

I think the authors have re-revised the manuscript almost adequately. I now support publication of this paper. Before publication, however, I would like to suggest the authors to improve the figure presentation as follows.

Looking at the newly added Supplementary Figure 3 using U2OS cells, I have realized that Treslin bands after immunoblots are not so clear in other figures mostly using HeLa cells. Could indicate which band(s) is (are) Treslin (upper, lower, or both band(s) ?) particularly in Figure 7e and 7b/HeLa.

1-Bands corresponding to Treslin have been indicated in Figures 7c (Fig 7b of the previous version), Figure 7h (Fig. 7d of the previous version), Figure 7j (Fig 7e of the previous version), Figures 9 a and b (previous Fig. 9a) and in Figures 9c, f and h (previous Figures 9b, d and e respectively).